# S-SOS: Stochastic Sum-Of-Squares for Parametric Polynomial Optimization

**Richard L. Zhu**
Department of Computational and Applied Mathematics
University of Chicago
Chicago, IL 60637
richardzhu@uchicago.edu

**Mathias Oster** *
Institute of Geometry and Practical Mathematics
RWTH Aachen
Aachen, Germany
oster@igpm.rwth-aachen.de

**Yuehaw Khoo**
Department of Statistics
University of Chicago
Chicago, IL 60637
yuehaw.khoo@uchicago.edu

## Abstract

Global polynomial optimization is an important tool across applied mathematics, with many applications in operations research, engineering, and the physical sciences. In various settings, the polynomials depend on external parameters that may be random. We discuss a stochastic sum-of-squares (S-SOS) algorithm based on the sum-of-squares hierarchy that constructs a series of semidefinite programs to jointly find strict lower bounds on the global minimum and extract candidates for parameterized global minimizers. We prove quantitative convergence of the hierarchy as the degree increases and use it to solve unconstrained and constrained polynomial optimization problems parameterized by random variables. By employing $n$-body priors from condensed matter physics to induce sparsity, we can use S-SOS to produce solutions and uncertainty intervals for sensor network localization problems containing up to 40 variables and semidefinite matrix sizes surpassing $800 \times 800$.

## 1 Introduction

Many effective nonlinear and nonconvex optimization techniques use local information to identify local minima. But it is often the case that we want to find global optima. Sum-of-squares (SOS) optimization is a powerful and general technique in this setting.

The core idea is as follows: suppose we are given polynomials $g_1, \ldots, g_m, f$ where each function is on $\mathbb{R}^n \to \mathbb{R}$ and we seek to determine the minimum value of $f$ on the closed set $\mathcal{S}$: $\mathcal{S} = \{x \in \mathbb{R}^n \mid g_i(x) \geq 0 \ \forall \ i = 1, \ldots, m\}$. Our optimization problem is then to find $\inf_{x \in \mathbb{R}^n} \{f(x) | x \in \mathcal{S}\}$.

---

*Corresponding Author

38th Conference on Neural Information Processing Systems (NeurIPS 2024).

An equivalent formulation is to find the largest constant $c \in \mathbb{R}$ (i.e. the tightest lower bound) that can be subtracted from $f$ such that $f - c \geq 0$ over the set $\mathcal{S}$. This reduction converts a polynomial optimization problem over a semialgebraic set to the problem of checking polynomial non-negativity. This problem is NP-hard in general [1], therefore one instead resorts to checking if $f - c$ is a sum-of-squares (SOS) function, e.g. in the unconstrained setting where $\mathcal{S} = \mathbb{R}^n$ one seeks to find some polynomials $h_k : \mathbb{R}^n \to \mathbb{R}$ such that $f - c = \sum_k h_k^2$. If such a decomposition can be found, then we have an easily-checkable certification that $f - c \geq 0$, as all sum-of-squares are non-negative. Since the converse is not true (not all non-negative functions are sum-of-squares), this is a relaxation of the original non-negativity problem.

Notably, if we restrict the $h_k$ to have maximum degree $d$, the search for a degree-$2d$ SOS decomposition of a function can be automated as a semidefinite program (SDP) [2, 3, 4]. Solving this SDP for varying degrees $d$ generates the well-known Lasserre (SOS) hierarchy. A given degree $d$ corresponds to a particular level of the hierarchy. Solving this SDP produces a lower bound $c_d$ which has been proven to converge to the true global minimum $c^* = \inf_x f(x)$ as $d$ increases, with finite convergence ($c_d = c^*$ at finite $d$) for functions with second-order local optimality conditions [5, 6] and asymptotic convergence with milder assumptions thanks to representation theorems for positive polynomials from real algebraic geometry [7, 8, 9]. Further work has elucidated both theoretical implications [7, 3, 10, 11, 12, 13] and useful applications of SOS to disparate fields [14, 15, 16, 5, 6, 17, 18] (see further discussion in Appendix A.2).

Motivated by the sum-of-squares certification for a lower bound $c$ on a function $f(x)$, we generalize to the case where the function to be minimized has additional parameters, i.e. $f(x, \omega)$ where $x$ are variables and $\omega$ are parameters drawn from some probability distribution $\omega \sim \nu(\omega)$. We seek a function $c(\omega)$ that is the tightest lower bound to $f(x, \omega)$ everywhere: $f(x, \omega) \geq c(\omega)$ with $c(\omega) \to \inf_x f(x, \omega)$. This setting was originally presented in [19] as a "Joint and Marginal" approach to parametric polynomial optimization. With the view that $\omega \sim \nu(\omega)$ and seeking to parameterize the minimizers $x^*(\omega) = \text{argmin}_x f(x, \omega)$, we are reminded of some of the prior work in polynomial chaos, where a system of stochastic variables is expanded into a deterministic function of those stochastic variables [20, 21].

**Contributions and outline.** Our primary contributions are a quantitative convergence proof for the Stochastic Sum-of-Squares (S-SOS) hierarchy of semidefinite programs (SDPs), a formulation of a new hierarchy (the cluster basis hierarchy) that uses the structure of a problem to sparsify the SDP, and numerical results on its application to the sensor network localization problem.

In Section 2, we review the S-SOS hierarchy of SDPs [19] and its primal and dual formulations (Section 2.1). We then detail how different hierarchies can be constructed (Section 2.2.1). Finally, in Section 2.3 (complete proof in Appendix A.5.2) we specialize to compact $X \times \Omega$ and outline the proof for a theorem on quantitative convergence (the gap between the optimal values of the degree-$2s$ S-SOS SDP and the "tightest lower-bounding" optimization problem goes $\to 0$ as $s \to \infty$) of the S-SOS hierarchy for trigonometric polynomials on $[0, 1]^n \times [0, 1]^d$ following the kernel formalism of [22, 6, 23].

In Section 3 we review the hierarchy's applications in parametric polynomial minimization and uncertainty quantification, focusing on several variants of sensor network localization on $X \times \Omega = [-1, 1]^n \times [-1, 1]^d$. We present numerical results for the accuracy of the extracted solutions that result from S-SOS, comparing to other approaches to parametric polynomial optimization, including a simple Monte Carlo-based method.

## 2 Stochastic Sum-of-squares (S-SOS)

### 2.0.1 Notation

Let $\mathcal{P}(S)$ be the space of polynomials on $S$, where $S \in \{X, \Omega\}$. $X \subseteq \mathbb{R}^n$ and $\Omega \subseteq \mathbb{R}^d$, respectively, where $X$ and $\Omega$ are (not-necessarily compact) subsets of their respective ambient spaces $\mathbb{R}^n$ and $\mathbb{R}^d$. A polynomial in $\mathcal{P}(X)$ can be written as $p(x) = \sum_{\alpha \in \mathbb{Z}_{\geq 0}^n} c_\alpha x^\alpha \in \mathcal{P}(X)$ (substituting $n \to d, x \to \omega, X \to \Omega$ for a polynomial in $\mathcal{P}(\Omega)$). Let $x := (x_1, \ldots, x_n), \omega := (\omega_1, \ldots, \omega_d)$, $\alpha$ be a multi-index (size given by context), and $c_\alpha$ be the polynomial coefficients. Let $\mathcal{P}^s(S)$ for some $s \in \mathbb{Z}_{\geq 0}, S \in \{X, \Omega\}$ denote the subspace of $\mathcal{P}(S)$ consisting of polynomials of degree $\leq s$, i.e. polynomials where the multi-indices of the monomial terms satisfy $||\alpha||_1 \leq s$. $\mathcal{P}_{\text{SOS}}(X \times \Omega)$ refers

to the space of polynomials on $X \times \Omega$ that can be expressible as a sum-of-squares in $x$ and $\omega$ jointly, and $\mathcal{P}_{\text{SOS}}^s(X \times \Omega)$ be the same space restricted to polynomials of degree $\leq s$. Additionally, $W \succeq 0$ for a matrix $W$ denotes that $W$ is symmetric positive semidefinite (PSD). Finally, $\mathbb{P}(\Omega)$ denotes the set of Lebesgue probability measures on $\Omega$. For more details, see Appendix A.1.

## 2.1 Formulation of S-SOS hierarchy

We present two formulations of the S-SOS hierarchy that are dual to each other in the sense of Fenchel duality [24, 25]. The primal problem seeks to find the tightest lower-bounding function and the dual problem seeks to find a minimizing probability distribution. Note that the "tightest lower bound" approach is dual to the "minimizing distribution" approach, otherwise known as a "joint and marginal" moment-based approach originally detailed in [19].

### 2.1.1 Primal S-SOS: The tightest lower-bounding function

Consider a polynomial $f(x, \omega) : \mathbb{R}^{n+d} \to \mathbb{R}$ with $x \in X \subseteq \mathbb{R}^n, \omega \in \Omega \subseteq \mathbb{R}^d$ equipped with a probability measure $\nu(\omega)$. We interpret $x$ as our optimization variables and $\omega$ as noise parameters, and seek a lower-bounding function $c^*(\omega)$ such that $f(x, \omega) \geq c^*(\omega)$ for all $x, \omega$. In particular, we want the tightest lower bound $c^*(\omega) = \inf_{x \in X} f(x, \omega)$. Note that even when $f(x, \omega)$ is polynomial, the tightest lower bound $c^*(\omega)$ can be non-polynomial. A simple example is the function $f(x, \omega) = (x - \omega)^2 + (\omega x)^2$, which has $c^*(\omega) = \inf_x f(x, \omega) = \omega^4/(1 + \omega^2)$ (Appendix A.6.1).

For us to select the "best" lower-bounding function, we want to maximize the expectation of the lower-bounding function $c(\omega)$ under $\omega \sim \nu(\omega)$ while requiring $f(x, \omega) - c(\omega) \geq 0$, giving us the following optimization problem over $L^1$-integrable lower-bounding functions:

$$p^* = \sup_{c \in L^1(\Omega)} \int c(\omega) \mathrm{d}\nu(\omega) \tag{1}$$
$$\text{s.t.} \quad f(x, \omega) - c(\omega) \geq 0$$

Even if we restricted $c(\omega)$ to be polynomial so that the residual $f(x, \omega) - c(\omega)$ is also polynomial, we would still have a challenging nonconvex optimization problem over non-negative polynomials. In SOS optimization, we take a relaxation and require the residual to be SOS: $f(x, \omega) - c(\omega) \in \mathcal{P}_{\text{SOS}}(X \times \Omega)$. Doing the SOS relaxation of the non-negative Equation (1) and restricting $c(\omega)$, i.e. $f(x, \omega) - c(\omega)$ to polynomials of degree $\leq 2s$ gives us Equation (2), which we call the primal S-SOS degree-$2s$ SDP:

$$p_{2s}^* = \sup_{c \in \mathcal{P}^{2s}(\Omega), W \succeq 0} \int c(\omega) \mathrm{d}\nu(\omega) \tag{2}$$
$$\text{s.t.} \quad f(x, \omega) - c(\omega) = m_s(x, \omega)^T W m_s(x, \omega)$$

where $m_s(x, \omega)$ is a basis function $X \times \Omega \to \mathbb{R}^{a(n,d,s)}$ containing monomial terms of degree $\leq s$ written as a column vector, and $W \in \mathbb{R}^{a(n,d,s) \times a(n,d,s)}$ a symmetric PSD matrix. Here, $a(n, d, s)$ represents the dimension of the basis function, which depends on the degree $s$ and on the dimensions $n, d$. For this formulation to find the best degree-$2s$ approximation to the lower-bounding function, we require $g(x, \omega) = m_s(x, \omega)^T W m_s(x, \omega)$ to span $\mathcal{P}^{2s}(X \times \Omega)$. Selecting all combinations of standard monomial terms of degree $\leq s$ suffices and results in a basis function with size $a(n, d, s) = \binom{n+d+s}{s}$.

### 2.1.2 Dual S-SOS: A minimizing distribution

The formal dual to Equation (1) (proof of duality in Appendix A.5.1) seeks to find a "minimizing distribution" $\mu(x, \omega)$, i.e. a probability distribution that places weight on the minimizers of $f(x, \omega)$ subject to the constraint that the marginal $\mu_X(\omega)$ matches $\nu(\omega)$:

$$d^* = \inf_{\mu \in \mathbb{P}(X \times \Omega)} \int f(x, \omega) \mathrm{d}\mu(x, \omega) \tag{3}$$
$$\text{s.t.} \quad \int_X \mathrm{d}\mu(x, \omega) = \mu_X(\omega) = \nu(\omega)$$

where we have written $\mathbb{P}(X \times \Omega)$ as the space of joint probability distributions on $X \times \Omega$ and $\mu_X(\omega)$ is the marginal of $\mu(x, \omega)$ with respect to $\omega$, obtained via disintegration.

For the primal, we considered polynomials of degree $\leq 2s$. We do the same here. The formal dual becomes a tractable SDP, where the objective turns into moment-minimization and the constraints become moment-matching. Following [3, 15], let $M \in \mathbb{R}^{a(n,d,s) \times a(n,d,s)}$ be the symmetric PSD moment matrix with entries defined as $M_{i,j} = \int_{X \times \Omega} m_s^{(i)}(x, \omega) m_s^{(j)}(x, \omega) d\mu(x, \omega)$ where $m_s^{(i)}(x, \omega)$ is the $i$-th element of the basis function $m_s$. Let $y \in \mathbb{R}^{b(n,d,s)}$ be the moment vector of independent moments that completely specifies $M$, e.g. in the case that we use all standard monomials of degree $\leq s$ and have $a(n, d, s) = \binom{n+d+s}{s}$, then $b(n, d, s) = \binom{n+d+2s}{2s}$. We write $M(y)$ as the moment matrix that is formed from these independent moments. We have $y_{\alpha(i,j)} = \int_{X \times \Omega} m_s^{(i)}(x, \omega) m_s^{(j)}(x, \omega) d\mu(x, \omega)$ where the multi-index $\alpha(i, j) \in \mathbb{Z}_{\geq 0}^{n+d}$ corresponds to the sum of the multi-indices corresponding to the $i$-th entry and the $j$-th entry of $m_s(x, \omega)$.

We write $f(x, \omega)$ in terms of the monomials $f(x, \omega) = \sum_{||\alpha||_1 \leq 2s} f_\alpha [x, \omega]^\alpha$, where $[x, \omega]$ is the concatenation of the $n + d$ variables from $x, \omega$ and $\alpha \in \mathbb{Z}_{\geq 0}^{n+d}$ is a multi-index. Note that every monomial $[x, \omega]^\alpha$ has a corresponding moment $y_\alpha$: $\int [x, \omega]^\alpha d\mu(x, \omega) = y_\alpha$. We then observe that the integral in the objective reduces to a dot product between the coefficients of $f$ and the moment vector:

$$\int f(x, \omega) d\mu(x, \omega) = \int \sum_\alpha f_\alpha [x, \omega]^\alpha d\mu(x, \omega) = \sum_\alpha f_\alpha y_\alpha$$

After converting the distribution-matching constraint $\mu_X(\omega) = \nu(\omega)$ in (3) into equality constraints on the moments of $\omega$ up to degree $2s$, we obtain the following dual S-SOS degree-$2s$ SDP:

$$d_{2s}^* = \inf_{y \in \mathbb{R}^{b(n,d,s)}} \sum_{||\alpha||_1 \leq 2s} f_\alpha y_\alpha \tag{4}$$
$$\text{s.t.} \quad M(y) \succcurlyeq 0$$
$$y_\alpha = m_\alpha \ \forall \ (\alpha, m_\alpha) \in \mathcal{M}_\nu$$

We write $\mathcal{M}_\nu$ as the set of $(\alpha, m_\alpha)$ representing the moment-matching constraints on $\omega^\alpha$ up to degree-$2s$, i.e. we want to set $\int_{X \times \Omega} \omega^\alpha d\mu(x, \omega) = \int_\Omega \omega^\alpha d\nu(\omega) = m_\alpha$ for all multi-indices $\alpha \in \mathbb{Z}_{\geq 0}^d$ with $||\alpha||_1 \leq 2s$. There are $\binom{d+2s}{2s}$ multi-indices $\alpha \in \mathbb{Z}_{\geq 0}^{n+d}, ||\alpha||_1 \leq 2s$ where only the $d$ entries associated with $\omega$ are non-zero, and therefore the number of moment-matching constraints is $|\mathcal{M}_\nu| = \binom{d+2s}{2s}$. Note that the moment matrix $M(y) \in \mathbb{R}^{a(n,d,s) \times a(n,d,s)}$ is a symmetric PSD matrix and is the dual variable to the primal $W$. Observe also that we require the moments of $\nu(\omega)$ of degree up to $2s$ to be bounded. (4) is often a more convenient form than (2), especially when working with additional equality or inequality constraints, as we will see in Section 3. For concrete examples of the primal and dual SDPs with explicit constraints, see Appendix A.3.

## 2.2 Variations

In this section, we detail two ways of building a hierarchy, one based on the maximum degree of monomial terms in the basis function (Lasserre) and a novel one based on the maximum number of interactions occurring in the terms of the basis function (cluster basis). To define any SOS hierarchy, we first select a monomial basis. Some examples include the standard monomial basis $x_1, \ldots, x_n$, trigonometric/Fourier 1-periodic monomial basis $\sin x_1, \cos x_1, \ldots, \sin x_n, \cos x_n$), or others. Using this basis, we write down a basis function $m(x)$ which comprises some combinations of monomials. Squared linear combinations of the basis functions then span a SOS space of functions: $\mathcal{H} : \{(\sum_i h_i m_i(x))^2\}$.

### 2.2.1 Standard Lasserre hierarchy

In the Lasserre hierarchy, the basis function $m_s(x)$ is composed of all combinations of monomials up to degree $s \in \mathbb{Z}_{>0}$ and a given level of the hierarchy is set by the maximum degree $s$. The basis function consists of terms $x^\alpha$ with $\alpha$ a multi-index and $||\alpha||_1 \leq s$. The degree-$2s$ SOS function space parameterized by this basis function is that spanned by $m_s(x)^T W m_s(x)$ for PSD $W$, i.e. the

functions that can result from squaring any linear combination of degree-$s$ polynomials that can be generated from our basis $m_s(x)$. As we increase the degree $s$, our basis function gets larger and our S-SOS SDP objective values converge to the optimal value of the "tightest lower-bounding" problem Equation (1) [19].

### 2.2.2 Cluster basis hierarchy

In this section, we propose a cluster basis hierarchy, wherein we utilize possible spatial organization of the problem to sparsify the problem and reduce the size of the SDP that must be solved [26, 27]. The cluster basis is a physically motivated prior often used in statistical and condensed matter physics, where we assume that our degrees of freedom can be arrayed in space, with locally close variables interacting strongly (kept in the model) and globally separated variables interacting weakly (ignored). Moreover, one may also keep only the terms with interactions between a small number of degrees of freedom, such as considering only pairwise or triplet interactions between particles.

In the cluster basis hierarchy, a given level of the hierarchy is specified by a 2-tuple $(b, t)$, the desired body order $b$ and the maximum degree of a single variable $t$. Body order denotes the maximum number of interacting variables in a given monomial term, e.g. $x_i^a x_j^b x_k^c$ would have body order 3 and total degree $a + b + c$. The basis function $m_{b,t}$ consists of terms $x^\alpha$ with $\alpha$ a multi-index, $||\alpha||_0 \leq b$ (at most $b$ interacting variables can occur in a single term), and $||\alpha||_\infty \leq t$ (each variable can have up to degree $t$. The maximum degree of the basis function $m_{b,t}$ is then $s = bt$. If we are to compare $m_{b,t}$ from the cluster basis hierarchy with $m_s$ from the Lasserre hierarchy, we find that even when $bt = s$ we still have strictly fewer terms, e.g. in the case where $b = 2, t = 2, s = 4$ we have $m_s$ containing terms of the form $x_i^4$ but $m_{b,t}$ only has degree-4 terms of the $x_i^2 x_j^2$.

To expand on this, consider that in the standard Lasserre hierarchy we have $m_s(x)$ containing all monomials of degree $\leq s$ in $n$ variables, or $\binom{n+s}{s}$ terms in total. In the proposed cluster basis hierarchy, $m_{b,t}(x)$ has

$$\sum_{k=0}^{b} \binom{n}{k} t^k$$

terms. This expression results from the need to sum over body orders $k$, considering that there are $\binom{n}{k}$ ways to choose $k$ variables and that each selected variable has $k$ possible degrees so there are $t^k$ ways to assign degrees $\leq t$ to these $k$ variables. For fixed $b, t$, $s = bt$, and $n \gg b, t$ we note that $m_s(x)$ has $O(n^{bt})$ terms while $m_{b,t}(x)$ has $O(n^b)$ terms so the size reduction factor in using the cluster basis asymptotically goes like $n^{b(t-1)}$. As the number of variables $n$ in the problem grow, we have asymptotic dominance in using the cluster basis to reduce the size of the SDP that must be solved. As $bt \to \infty$ we might expect asymptotic convergence of the SDP hierarchy just like the standard Lasserre hierarchy, however, a full proof of convergence is out of scope for this paper. For further details, see discussion in Appendix A.7.4.

### 2.3 Convergence of S-SOS

As we increase the degree $s$ (either $s$ in the Lasserre hierarchy or $b, t$ in the cluster basis hierarchy) we would expect the SDP objective values $p_{2s}^*$ (Equation (2)) to converge to the optimal value $p^*$ and the lower bounding function $c_{2s}^*(\omega)$ to converge to the tightest lower bound $c^*(\omega) = \inf_x f(x, \omega)$. In this paper we refer to $p_{2s}^* \to p^*$ and $d_{2s}^* \to d^*$ interchangeably as strong duality occurs in practice despite being difficult to formally verify (Appendix A.4). This convergence is a common feature of SOS hierarchies.

In this section we show that using polynomial $c_{2s}^*(\omega)$ to approximate $c^*(\omega)$ still allows for asymptotic convergence in $L^1$ as $s \to \infty$.

### 2.3.1 Overview of result

We specialize to the particular case of 1-periodic trigonometric polynomials $f(x, \omega), c(\omega)$ on compact $X = [0, 1]^n$ and compact $\Omega \subset \mathbb{R}^d$ and prove asymptotic convergence of the degree-$2s$ S-SOS hierarchy as $s \to \infty$. Though seemingly restrictive, this choice allows us to leverage Fourier convergence results on a compact domain. For generic $f(x, \omega)$ where we are only interested in its behavior in some compact set (nearly all practical problems involve a restriction of domain), we may

rescale the compact set to this 1-periodic compact domain and apply the results found here. To use other families of polynomials, we note that a substitution argument justifies the focus on trigonometric polynomials as any convergence result achieved for the trigonometric polynomial hierarchy lead directly to a matching result on regular polynomials (2.2 in [28]). Previously, rates of $1/s^2$ were attained for the standard Lasserre hierarchy (i.e. without external parameters $\omega$) on compact domains with mild assumptions [29, 28, 9]. In our work, a slower convergence rate of $\ln s/s$ is achieved due to the introduction of external parameters, requiring us to approximate the lower-bounding function $c^*(\omega)$ and thereby limiting the obtained convergence rate.

### 2.3.2 $\ln s/s$ convergence using a polynomial approximation to $c^*(\omega)$

We would like to bound the gap between the optimal lower bound $c^*(\omega) = \inf_{x \in X} f(x, \omega)$ and the lower bound $c_{2s}^*(\omega)$ resulting from solving the degree-$2s$ primal S-SOS SDP, i.e.

$$0 \le c^*(\omega) - c_{2s}^*(\omega) \le \varepsilon(f, s) \ \forall \ \omega \in \Omega. \tag{5}$$

To that end, we need to understand the regularity of $c^*$. Without further assumptions, we may assume $c^*$ to be Lipschitz continuous, per Proposition 2.1.

With Equation (5) we may then integrate

$$0 \le \int_\Omega \inf_x f(x, \omega) - c_{2s}^*(\omega) \mathrm{d}\nu(\omega) \le |\Omega| \varepsilon(f, s)$$

where we control $\varepsilon$ in terms of the degree $s$. If we can drive $\epsilon \to 0$ as $s \to \infty$ then we are done.

**Proposition 2.1** (Theorem 2.1 in [30]). *Let $g : X \times Y \to \mathbb{R}$ be polynomial. Then $y \mapsto \inf_{x \in X} g(x, y)$ is Lipschitz continuous.*

**Theorem 2.1** (Asymptotic convergence of S-SOS). *Let $f : [0, 1]^n \times \Omega \to \mathbb{R}$ be a trigonometric polynomial of degree $2r$, $c^*(\omega) = \inf_x f(x, \omega)$ the optimal lower bound as a function of $\omega$, and $\nu$ any probability measure on compact $\Omega \subset \mathbb{R}^d$. Let $s$ refer to the degree of the basis in both $x, \omega$ terms and the degree of the lower-bounding polynomial $c(\omega)$, i.e. $m_s([x, \omega]) : \mathbb{R}^{n+d} \to \mathbb{R}^{a(n,d,s)}$ is the full basis function of terms $[x, \omega]^\alpha$ with $||\alpha||_1 \le s$ and $c(\omega)$ only has terms $\omega^\alpha$ with $||\alpha||_1 \le s$.*

*Let $p_{2s}^*$ be the solution to the following S-SOS SDP (c.f. Equation (2)) with $m_s(x, \omega)$ a spanning basis of trigonometric monomials with degree $\le s$:*

$$p_{2s}^* = \sup_{c \in \mathcal{P}^{2s}(\Omega), W \succcurlyeq 0} \int c(\omega) \mathrm{d}\nu(\omega)$$
$$s.t. \quad f(x, \omega) - c(\omega) = m_s(x, \omega)^T W m_s(x, \omega)$$

*Then there is a constant $C > 0$ depending only on $||f - \bar{f}||_F, ||c^* - \bar{c}^*||, r, \Omega, n, d$ such that the following holds:*

$$\int_\Omega [c^*(\omega) - c_{2s}^*(\omega)] \, \mathrm{d}\nu(\omega) \le C \frac{\ln s}{s}$$

*where $\bar{f}$ denotes the average value of the function $f$ over $[0, 1]^n$, i.e. $\bar{f} = \int_{[0,1]^n} f(x) dx$ and $||f(x)||_F = \sum_{\hat{x}} |\hat{f}(\hat{x})|$ denotes the norm of the Fourier coefficients. Thus we have asymptotic convergence of the S-SOS SDP hierarchy to the optimal value $p^*$ of Equation (1) as we send $s \to \infty$.*

*Proof.* The following is an outline of the proof. For complete details, including the full theorem and proof, please see Appendix A.5.2.

The key argument used follows that of [22, 28, 23, 31, 32]. Take the lower-bounding minimizer of Equation (1) $c^*(\omega)$ (not necessarily polynomial) and approximate it with a trigonometric polynomial $c_a^*(\omega)$ of degree $s$. We then pass the non-negative component $f - c_a^*$ through an integral operator $T$ that is built out of sums of squares of trigonometric polynomials (related to the Christoffel-Darboux and Jackson kernels) to obtain a strictly-positive SOS component:

$$Th(x, \omega) = \int_{X \times \Omega} |q_x(x - \bar{x})|^2 |q_\omega(\omega - \bar{\omega})|^2 h(\bar{x}, \bar{\omega}) \mathrm{d}\bar{x} \mathrm{d}\bar{\omega}.$$

$q_x(x), q_\omega(\omega)$ can be chosen to be "kernel functions" of bounded degree $\leq s$ so that the output SOS function is of degree $2s$. We can show the approximation error $||c^* - c_a^*||$ is small, that the operator $T$ exists and is close to the identity, and the deformation resulting from the SOS projection is small for sufficiently large degree $s$ (degree of the approximating $c_a^*(\omega)$ and in the SOS kernel). We may conclude that the true strictly-positive part may be well-approximated by the SOS hierarchy and find asymptotic convergence along with a convergence rate in the degree $s$ of the hierarchy.

$\square$

## 3 Numerical experiments

We present two numerical studies of S-SOS demonstrating its use in applications. The first study (Section 3.1) numerically tests how the optimal values of the SDP Equation (2) $p_{2s}^*$ converge to $p^*$ of the original primal Equation (1) as we increase the degree. The second study (Section 3.2) evaluates the performance of S-SOS for solution extraction and uncertainty quantification in various sensor network localization problems.

### 3.1 Simple quadratic SOS function

As a simple illustration of S-SOS, we test it on the SOS function

$$f(x, \omega) = (x - \omega)^2 + (\omega x)^2 \tag{6}$$

with $x \in \mathbb{R}, \omega \in \mathbb{R}$. The lower bound $c^*(\omega) = \inf_x f(x, \omega)$ can be computed analytically as $c^*(\omega) = \omega^4/(1 + \omega^2)$. Assuming $\omega \sim \text{Uniform}(-1, 1)$, we get that the objective value for the "tightest lower-bounding" primal problem Equation (1) is $p^* = \int_{-1}^1 \frac{\omega^4}{2(1+\omega^2)} d\omega = \frac{\pi}{4} - \frac{2}{3} \approx 0.1187$. For further details, see Appendix A.6.

We are interested in studying the quantitative convergence of the S-SOS hierarchy numerically. The idea is to solve the primal (dual) degree-$2s$ SDP to find the tightest polynomial lower bound (the minimizing probability distribution) for varying degrees $s$. As $s$ gets larger, the basis function $m_s(x)$ gets larger and the objective value of the SDP Equation (2) $p_{2s}^*$ should converge to the theoretical optimal value $p^*$.

In Figure 1 we see very good agreement between $p^*$ and $p_{2s}^*$ with exponential convergence as $s$ increases. This is much faster than the rate we found in Section 2.3.2, but agrees with the exponential convergence results from [6] achieved with local optimality assumptions. Due to the simplicity of (6), it is not surprising that we see much faster convergence. In fact, for most typical functions, we might expect convergence much faster than the worst-case rate. The tapering-off of the convergence rate is likely attributed to the numerical tolerance used in our solver (CVXPY/MOSEK), as we observed that increasing the tolerance shifts the best-achieved gap higher.

### 3.2 Sensor network localization

Sensor network localization (SNL) is a common testbed for global optimization and SDP solvers due to the high sensitivity and ill-conditioning of the problem. In SNL, one seeks to recover the positions of $N$ sensors $X \in \mathbb{R}^{N \times \ell}$ positioned in $\mathbb{R}^\ell$ given a set of noisy observations of pairwise distances $d_{ij} = ||x_i - x_j||$ between the sensors [15, 33]. To have a unique global minimum and remove symmetries, sensor-anchor distance observations are often added, where several sensors are anchored at known locations in the space. This can improve the conditioning of the problem, making it "easier" in some sense.

#### 3.2.1 Definitions

We define a SNL *problem instance* with $X \in [-1, 1]^{N \times \ell}$ as the ground-truth positions for $\mathcal{S} = \{1, 2, \ldots, N\}$ sensors, $A \in [-1, 1]^{K \times \ell}$ as the ground-truth positions for $\mathcal{A} = \{1, 2, \ldots, K\}$ anchors, $\mathcal{D}_{ss}(r) = \{d_{ij} = ||x_i - x_j|| : i, j \in \mathcal{S} \text{ and } d_{ij} \leq r\}$ as the set of observed sensor-sensor distances and $\mathcal{D}_{sa}(r) = \{d_{ik} = ||x_i - a_k|| : i \in \mathcal{S}, k \in \mathcal{A} \text{ and } d_{ik} \leq r\}$ as the set of observed sensor-anchor distances, both of which depend on some sensing radius $r$.

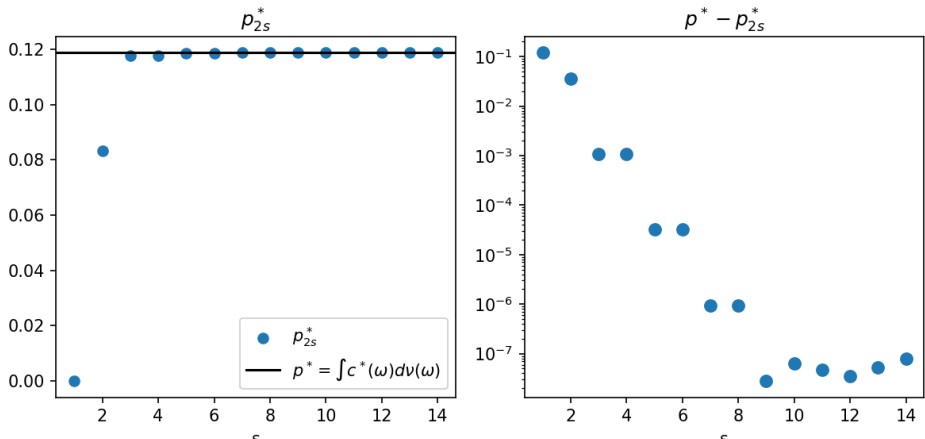

Figure 1: Comparison between the objective value $p_{2s}^*$ from solving the degree-$2s$ S-SOS SDP and the objective value $p^*$ resulting from the best-possible lower bound $c^*(\omega)$ for noise drawn as $\omega \sim \text{Uniform}(-1, 1)$. $p^* = \int c^*(\omega)d\nu(\omega) = \frac{\pi}{4} - \frac{2}{3} \approx 0.1187$ is plotted as the line in black and the $p_{2s}^*$ values are shown as blue dots (left) with the gap between the values $p^* - p_{2s}^*$ (right).

Writing $x_i, a_k \in [-1, 1]^\ell$ as the unknown positions of the $i$-th sensor and the $k$-th anchor, we can write the potential function to be minimized as a polynomial:

$$f(x, \omega; X, A, r) = \underbrace{\sum_{d_{ij} \in \mathcal{D}_{ss}(r)} (\|x_i - x_j\|_2^2 - d_{ij}(\omega)^2)^2}_{\text{sensor-sensor interactions}} + \underbrace{\sum_{d_{ik} \in \mathcal{D}_{sa}(r)} (\|x_i - a_k\|_2^2 - d_{ik}(\omega)^2)^2}_{\text{sensor-anchor interactions}} \quad (7)$$

The observed sensor-sensor and sensor-anchor distances $d_{ij}(\omega), d_{ik}(\omega)$ can be perturbed arbitrarily, but in this paper we focus on linear uniform noise, i.e. for a subset of observed distances we have $d_{ij,k}(\omega) = d_{ij}^* + \epsilon\omega_k$ with $\omega_k \sim \text{Uniform}(-1, 1)$. Other noise types may be explored, including those including outliers, which may be a better fit for robust methods (Appendix A.7.2).

Equation (7) contains soft penalty terms for sensor-sensor terms and sensor-anchor terms. We can see that this is a degree-4 polynomial in the standard monomial basis elements, and a global minimum of this function is achieved at $f(X, \mathbf{0}^d; X, A, r) = 0$ (where the distances have not been perturbed by any noise). In general for non-zero $\omega$ (measuring distances under noise perturbations) we expect the function minimum to be $> 0$, as there may not exist a configuration of sensors $\hat{X}$ that is consistent with the observed noisy distances.

We can also support equality constraints in our solution, in particular hard equality constraints on the positions of certain sensors relative to known anchors. This corresponds to removing all sensor-anchor soft penalty terms from the function and instead selecting $N_H < N$ sensors at random to exactly fix in known positions via equality constraints in the SDP. The SDP is still large but the effective number of variable sensors has been reduced to $N' = N - N_H$.

A given SNL *problem type* is specified by a spatial dimension $\ell$, $N$ sensors, $K$ anchors, a sensing radius $r \in (0, 2\sqrt{\ell})$, a noise type (linear), and anchor type (soft penalty or hard equality). Once these are specified, we generate a random *problem instance* by sampling $X \sim \text{Uniform}(-1, 1)^n, A \sim \text{Uniform}(-1, 1)^d$. The potential $f(x, \omega)$ for a given instance is formed (either with sensor-anchor terms or not, with terms kept based on some sensing radius $r$, and noise variables appropriately added).

The number of anchors is chosen to be as few as possible so as to still enable exact localization, i.e. $K = \ell + 1$ anchors for a SNL problem in $\ell$ spatial dimensions. The SDPs are formulated with the help of SymPy [34] and solved using CVXPY [35, 36] and Mosek [37] on a server with two Intel Xeon 6130 Gold processors (32 physical cores total) and 256GB of RAM. For an expanded discussion and further details, see Appendix A.7.

### 3.2.2 Evaluation metrics

The accuracy of the recovered solution is of primary interest, i.e. our primary evaluation metric should be the distance between our extracted sensor positions $x$ and the ground-truth sensor positions $X$, i.e. dist$(x, X)$. Because the S-SOS hierarchy recovers estimates of the sensor positions $\mathbb{E}[x_i]$ along with uncertainty estimates $\mathrm{Var}[x_i]$, we would like to measure the distance between our ground-truth positions $X$ to our estimated distribution $p(x) = \mathcal{N}(\mathbb{E}[x], \mathrm{Var}[x])$. The Mahalanobis distance $\delta_M$ (Equation (8)) is a modified distance metric that accounts for the uncertainty [38]. We use this as our primary metric for sensor recovery accuracy.

$$\delta_M(X, \mathcal{N}(\mu, \Sigma)) := \sqrt{(X - \mu)^T \Sigma^{-1} (X - \mu)} \tag{8}$$

As our baseline method, for each problem instance we apply a basic Monte Carlo method detailed in Algorithm 1 (Appendix A.7.3) where we sample $\omega \sim \nu(\omega)$, use a local optimization solver to find $x^*(\omega) = \inf_x f(x, \omega)$, and use this to estimate $\mathbb{E}_{\omega \sim \nu}[x], \mathrm{Var}_{\omega \sim \nu}[x]$. Note that though this non-SOS method achieves some estimate of the dual SDP objective $\int f(x, \omega) d\mu(x, \omega)$, it is not guaranteed to be a lower bound.

### 3.2.3 Results

**Recovery accuracy.** In Table 1 we see a comparison of the S-SOS method and the MCPO baseline. Each row corresponds to one SNL problem type, i.e. we fix the physical dimension $\ell$, the number of anchors $K = \ell + 1$, and select the sensing radius $r$ and the noise scale $\epsilon$. We then generate $L = 20$ random instances of each problem type, corresponding to a random realization of the ground-truth sensor and anchor configurations $X \in [-1, 1]^{N \times \ell}, A \in [-1, 1]^{K \times \ell}$, producing a $f(x, \omega)$ that we then solve the SDP for (in the case of S-SOS) or do pointwise optimizations for (in the case of MCPO). Each method outputs estimates for the sensor positions and uncertainty around it as a $\mathcal{N}(\mathbb{E}[x], \mathrm{Cov}[x])$, which we then compute $\delta_M$ for (see Equation (8)), treating each dimension as independent of each other (i.e. $X$ as a flat vector). Each instance solve gives us one observation of $\delta_M$ or each method, and we report the median and the $\pm 1\sigma_{34\%}$ values over the $L = 20$ instances we generate.

## 4 Discussion

In this paper, we discuss the stochastic sum-of-squares (S-SOS) method to solve global polynomial optimization in the presence of noise, prove two asymptotic convergence results for polynomial $f$ and compact $\Omega$, and demonstrate its application to parametric polynomial minimization and uncertainty quantification along with a new cluster basis hierarchy that enables S-SOS to scale to larger problems. In our experiments, we specialized to sensor network localization and low-dimensional uniform random noise with small $n, d$. However, it is relatively straightforward to extend this method to support other noise types (such as Gaussian random variates without compact support, which we do in Appendix A.6.4) and support higher-dimensional noise with $d \gg 1$.

Scaling this method to larger problems $n \gg 1$ is an open problem for all SOS-type methods. In this paper, we take the approach of sparsification, by making the cluster basis assumption to build up a block-sparse $W$. We anticipate that methods that leverage sparsity or other structure in $f$ will be promising avenues of research, as well as approximate solving methods that avoid the explicit materialization of the matrices $W, M$. For example, we assume that the ground-truth polynomial possesses the block-sparse structure because our SDP explicitly requires the polynomial $f(x, \omega)$ to exactly decompose into some lower-bounding $c(\omega)$ and SOS $f_{\mathrm{SOS}}(x, \omega)$. Relaxing this exact-decomposition assumption and generalizing beyond polynomial $f(x, \omega), c(\omega)$ may require novel approaches and would be an exciting area for future work.

Table 1: Comparison of S-SOS and MCPO solution extraction accuracy. We present the Mahalanobis distance $\delta_M$ (Equation (8)) of the the true sensor positions $X^*$ to the extracted distribution $\mathcal{N}(\mathbb{E}[x], \text{Var}[x])$ over solutions recovered from S-SOS for varying SNL problem types. $\ell$ is the spatial dimension, $r$ is the sensing radius used to cutoff terms in the potential $f(x, \omega)$, $\epsilon$ is the noise scale, $N_H$ is the number of hard equality constraints used (sensors fixed at known locations), $N_C$ is the number of clusters used (see Appendix A.7.4), and $N$ is the number of sensors used. Each SNL problem instance has $K = \ell + 1$ anchors used in the potential (if $N_H = 0$). The MCPO values are estimated with $T = 50$ Monte Carlo iterates. Each entry is $\hat{\mu} \pm \hat{\sigma}$ where $\hat{\mu}$ is the median and robust standard-deviation ($\sigma_{34\%}$) estimated over 20 runs of the same problem type with varying random initializations of the sensor positions. The entries with the lowest median $\delta_M$ are bolded. We also compare the number of elements in the full basis $a_f$, the cluster basis $a_c$, and the reduction multiple when using the cluster basis $a_f/a_c$. When passing to the cluster basis, $a_f/a_c$ is how much the semidefinite matrix shrinks by.

| Parameters | | | | | | Basis comparison | | | M-distance ($\delta_M$) | |
|---|---|---|---|---|---|---|---|---|---|---|
| $\ell$ | $r$ | $\epsilon$ | $N_H$ | $N_C$ | $N$ | $a_f$ | $a_c$ | $a_f/a_c$ | S-SOS | MCPO |
| 1 | 0.5 | 0.3 | 0 | 1 | 10 | 78 | 78 | 1x | $\mathbf{0.94 \pm 0.22}$ | $2.61 \pm 3.86$ |
| 1 | 1.0 | 0.3 | 0 | 1 | 10 | 78 | 78 | 1x | $\mathbf{0.29 \pm 0.16}$ | $1.10 \pm 0.58$ |
| 1 | 1.5 | 0.3 | 0 | 1 | 10 | 78 | 78 | 1x | $\mathbf{0.11 \pm 0.11}$ | $0.86 \pm 0.52$ |
| 1 | 1.5 | 0.3 | 2 | 1 | 10 | 78 | 78 | 1x | $\mathbf{0.24 \pm 0.37}$ | $1.06 \pm 1.28$ |
| 1 | 1.5 | 0.3 | 4 | 1 | 10 | 78 | 78 | 1x | $\mathbf{0.10 \pm 0.03}$ | $0.61 \pm 0.41$ |
| 1 | 1.5 | 0.3 | 6 | 1 | 10 | 78 | 78 | 1x | $\mathbf{0.06 \pm 0.04}$ | $0.48 \pm 0.32$ |
| 1 | 1.5 | 0.3 | 8 | 1 | 10 | 78 | 78 | 1x | $\mathbf{0.04 \pm 0.02}$ | $0.31 \pm 0.17$ |
| 2 | 1.5 | 0.1 | 0 | 9 | 9 | 406 | 163 | 2.5x | $\mathbf{2.86 \pm 0.94}$ | $1562.39 \pm 596.29$ |
| 2 | 1.5 | 0.1 | 0 | 9 | 15 | 820 | 317 | 2.6x | $\mathbf{3.25 \pm 1.19}$ | $1848.65 \pm 650.45$ |

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

# A  Appendix / supplemental material

## A.1  Notation

Let $\mathcal{P}(X)$ and $\mathcal{P}(\Omega)$ denote the spaces of polynomials on $X \subseteq \mathbb{R}^n$ and $\Omega \subseteq \mathbb{R}^d$, respectively, where $X$ and $\Omega$ are (not-necessarily compact) subsets of their respective ambient spaces $\mathbb{R}^n$ and $\mathbb{R}^d$. Specifically, all polynomials of the forms below belong to their respective spaces:

$$p(x) = \sum_{\alpha \in \mathbb{Z}_{\geq 0}} c_\alpha x^\alpha \in \mathcal{P}(X), \quad p(\omega) = \sum_{\alpha \in \mathbb{Z}_{\geq 0}} c_\alpha \omega^\alpha \in \mathcal{P}(\Omega)$$

where $x = (x_1, \ldots, x_n), \omega = (\omega_1, \ldots, \omega_d)$, $\alpha$ is a multi-index for the respective spaces, and $c_\alpha$ are the polynomial coefficients.

Let $\mathcal{P}^d(S)$ for some $S \in \{X, \Omega\}$ denote the subspace of $\mathcal{P}(S)$ consisting of polynomials of degree $\leq d$, i.e. polynomials where the multi-indices of the monomial terms satisfy $||\alpha||_1 \leq d$. $\mathcal{P}_{\text{SOS}}(X \times \Omega)$ refers to the space of polynomials on $X \times \Omega$ that can be expressible as a sum-of-squares in $x$ and $\omega$ jointly. Additionally, $W \succeq 0$ for a matrix $W$ denotes that $W$ is symmetric positive semidefinite (PSD). Finally, $\mathbb{P}(\Omega)$ denotes the set of Lebesgue probability measures on $\Omega$.

## A.2  Related work

### A.2.1  Sum-of-squares theory and practice

The theoretical justification underlying the SDP relaxations in global optimization we use here derive from the Positivstellensätz (positivity certificate) of [7], a representation theorem guaranteeing that strictly positive polynomials on certain sets admit sum-of-squares representations. Following this, [3, 10, 11] developed the Moment-SOS hierarchy, describing a hierarchy of primal-dual SDPs (each having fixed degree) of increasing size that provides a monotonic non-decreasing sequence of lower bounds.

There is rich theory underlying the SOS hierarchy combining disparate results from algebraic geometry [14, 10, 11], semidefinite programming [15, 18], and complexity theory [16, 39]. The hierarchy exhibits finite convergence in particular cases where convexity and a strict local minimum are guaranteed [5], otherwise converging asymptotically [6]. In practice, the hierarchy often does even better than these guarantees, converging exactly at $c_s^*$ for some small $s$.

The SOS hierarchy has found numerous applications in wide-ranging fields, including: reproducing certain results of perturbation theory and providing useful lower-bound certifications in quantum field theory and quantum chemistry [40, 41], providing better provable guarantees in high-dimensional statistical problems [42, 43], useful applications in the theory and practice of sensor network localization [15, 44] and in robust and stochastic optimization [45].

Due to the SDP relaxation, the SOS hierarchy is quite powerful. This flexibility comes at a cost, primarily in the form of computational complexity. The SDP prominently features a PSD matrix $W \in \mathbb{R}^{a(n,d,s) \times a(n,d,s)}$ with $a(n, d, s)$ scaling as $\binom{n+d+s}{s}$ for $n$ dimensions and maximum degree $s$. Without exploiting the structure of the polynomial, such as locality (coupled terms) or sparsity, solving the SDP using a standard interior point method becomes prohibitively expensive for moderate values of $s$ or $n$. Work attempting to improve the scalability of the core ideas underlying the SOS hierarchy and the SDP method include [17, 18].

### A.2.2  Stochastic sum-of-squares and parametric polynomial optimization

The S-SOS hierarchy we present in this paper as a solution to parametric polynomial optimization was presented originally by [19] as a "Joint + Marginal" approach. That work provides the same hierarchy of semidefinite relaxations where the sequence of optimal solutions converges to the moment vector of a probability measure encoding all information about the globally-optimal solutions $x^*(\omega) = \text{argmin}_x f(x, \omega)$ and provides a proof that the dual problem (our primal) obtains a polynomial approximation to the optimal value function that converges almost-uniformly to $c^*(\omega)$.

### A.2.3 Uncertainty quantification and polynomial chaos

Once a physical system or optimization problem is characterized, sensitivity analysis and uncertainty quantification seek to quantify how randomness or uncertainty in the inputs can affect the response. In our work, we have the parametric problem of minimizing a function $f(x, \omega)$ over $x$ where $\omega$ parameterizes the function and is drawn from some noise distribution $\nu(\omega)$.

If only function evaluations $f(x, \omega)$ are allowed and no other information is known, Monte Carlo is often applied, where one draws $\omega_k \sim \nu(\omega)$ and solves many realizations of $\inf_x f_k(x) = f(x, \omega_k)$ to approximately solve the following stochastic program:

$$f^* = \inf_x \mathbb{E}_{\omega \sim \nu}[f(x, \omega)]$$

Standard Monte Carlo methods are ill-suited for integrating high-dimensional functions, so this method is computationally challenging in its own right. In addition, we have no guarantees on our result except that as we take the number of Monte Carlo iterates $T \to \infty$ we converge to some unbiased estimate of $\mathbb{E}_{\omega \sim \nu}[f(x, \omega)]$.

Our approach to quantifying the uncertainty in optimal function value resulting from uncertainty in parameters $\omega$ is to find a deterministic lower-bounding $c^*(\omega)$ which guarantees $f(x, \omega) \geq c^*(\omega)$ no matter the realization of noise. This is reminiscent of the polynomial chaos expansion literature, wherein a system of some stochastic variables is expanded into a deterministic function of those stochastic variables, usually in some orthogonal polynomial basis [20, 21].

### A.3 An example

**Example A.1.** Let $f(x, \omega)$ be some polynomial of degree $\leq 2s$ written in the standard monomial basis, i.e.

$$f(x, \omega) = \sum_{||\alpha||_1 \leq 2s} f_\alpha x^\alpha$$

$$= \sum_{||\alpha||_1 \leq 2s} f_{(\alpha_1, \ldots, \alpha_{n+d})} \prod_{i=1}^n x_i^{\alpha_i} \prod_{i=1}^d \omega_i^{\alpha_{n+i}}$$

Let $m_s(x, \omega) \in \mathbb{R}^{a(n,d,s)}$ be the basis vector representing the full set of monomials in $x, \omega$ of degree $\leq s$ with $a(n, d, s) = \binom{n+d+s}{s}$.

For all $\alpha \in \mathbb{Z}_{\geq 0}^{n+d}$ with $||\alpha||_1 \leq 2s$ and $\alpha_k = 0$ for all $k \in \{1, \ldots, n\}$ (i.e. monomial terms containing only $\omega_1, \ldots, \omega_d$) we must have:

$$\int_{X \times \Omega} \omega^\alpha d\mu(x, \omega) - \int_\Omega \omega^\alpha d\nu(\omega) = 0$$

Explicitly, for $\mu$ to be a valid probability distribution we must have:

$$\int_{X \times \Omega} d\mu(x, \omega) - 1 = M_{0,0} - 1 = y_{(1,0,\ldots)} - 1 = 0$$

Suppose $\Omega = [-1, 1], \omega \sim \text{Uniform}(-1, 1)$ so that $d = 1, \nu(\omega) = 1/2$. We require:

$$\int_{X \times \Omega} \omega^\alpha d\mu(x, \omega) = \int_{[-1,1]} \omega^\alpha d\nu(\omega) = \begin{cases} 1 & \alpha = 0 \\ 0 & \alpha = 1 \\ \frac{1}{3} & \alpha = 2 \\ 0 & \alpha = 3 \\ \frac{1}{5} & \alpha = 4 \end{cases}$$

$\square$

## A.4 Strong duality

To guarantee strong duality theoretically, we need a strictly feasible point in the interior (Slater's condition). For us, this is a consequence of Putinar's Positivstellensatz, if $f(x, \omega)$ admits a decomposition as $f(x, \omega) = c(\omega) + g(x, \omega)$ where $g(x, \omega) > 0$ (i.e. is strictly positive), we have strong duality, i.e. $p^* = d^*$ and $p_{2s}^* = d_{2s}^*$ [3, 8]. However, it is difficult to verify the conditions analytically. In practice, strong duality is observed in most cases, so in this paper we refer to solving the primal and dual interchangeably, as $p_{2s}^* = d_{2s}^*$ in all cases we encounter where a SDP solver returns a feasible point.

## A.5 Proofs

### A.5.1 Primal-dual relationship of S-SOS

**Regular SOS**  Global polynomial optimization can be framed as the following lower-bound maximization problem where we need to check global non-negativity:

$$\sup_{c \in \mathbb{R}} \quad c \tag{9}$$
$$\text{s.t.} \quad f(x) - c \geq 0 \quad \forall x$$

When we take the SOS relaxation of the non-negativity constraint in the primal, we now arrive at the SOS primal problem, where we require $f(x) - c$ to be SOS which guarantees non-negativity but is a stronger condition than necessary:

$$\sup_{c \in \mathbb{R}} \quad c \tag{10}$$
$$\text{s.t.} \quad f(x) - c \in \mathcal{P}_{\text{SOS}}(X).$$

The dual to Equation (9) is the following moment-minimization problem:

$$\inf_{\mu \in \mathbb{P}(X)} \quad \int f(x) \mathrm{d}\mu(x) \tag{11}$$
$$\text{with} \quad \int \mathrm{d}\mu(x) = 1.$$

Taking some spanning basis $m_s(x) : \mathbb{R}^n \to \mathbb{R}^{a(n,s)}$ of monomials up to degree $s$, we have the moment matrix $M \in \mathbb{R}^{a(n,s) \times a(n,s)}$:

$$M_{i,j} = \int m_i(x) m_j(x) d\mu(x) = y_\alpha$$

where we introduce a moment vector $y$ whose elements correspond to the unique moments of the matrix $M$. Then we may write the degree-$2s$ moment-minimization problem, which is now in a solvable numerical form:

$$\inf_{y} \quad \sum_{\alpha} f_\alpha y_\alpha \tag{12}$$
$$\text{with} \quad M(y)_{1,1} = 1$$
$$M(y) \succcurlyeq 0$$

where we write $M(y)$ as the matrix formed by placing the moments from $y$ into their appropriate places and we set the first element of $m_s(x)$ to be 1, hence $M_{1,1} = \int d\nu(x) = 1$ is simply the normalization constraint. For further reading, see [15, 3].

**Stochastic SOS**  Now let us lift this problem into the stochastic setting with parameters $\omega$ sampled from a given distribution $\nu$, i.e. replacing $x \to (x, \omega)$. We need to make some choice for the objective. The expectation of the lower bound under $\nu(\omega)$ is a reasonable choice, i.e.

$$\int_\Omega c(\omega) d\nu(\omega)$$

but we could also make other choices, such as ones that encourage more robust lower bounds. In this paper however, we formulate the primal S-SOS as below (same as Equation (1)):

$$p^* = \sup_{c \in L^1(\Omega)} \int c(\omega)\mathrm{d}\nu(\omega) \tag{13}$$

$$\text{s.t.} \quad f(x,\omega) - c(\omega) \geq 0$$

Note that if the ansatz space for the function $c(\omega)$ is general enough, the maximization of the curve $c$ is equivalent to a pointwise maximization, i.e. we recover the best approximation for almost all $\omega$. Then the dual problem has a very similar form to the non-stochastic case.

**Theorem A.2.** *The dual to Equation* (13) *is the following moment minimization where* $\mu(x,\omega)$ *is a probability measure on* $X \times \Omega$:

$$\inf_{\mu \in \mathbb{P}(X \times \Omega)} \int f(x,\omega)\mathrm{d}\mu(x,\omega)$$

$$\text{with} \quad \int_{X \times \Omega} \omega^\alpha \mathrm{d}\mu(x,\omega) = \int_\Omega \omega^\alpha \mathrm{d}\nu(\omega) \quad \text{for all } \alpha \in \mathbb{N}^d.$$

*Remark* A.3. Notice, that the condition $\int_{X \times \Omega} \omega^\alpha \mathrm{d}\mu(x,\omega) = \int_\Omega \omega^\alpha \mathrm{d}\nu(\omega)$ implies that the first marginal of $\mu$ is the noise distribution $\nu$. Let $\mu_\omega$ denote the disintegration of $\mu$ with respect to $\nu$, [46]. Then the moment matching condition is equivalent to $\mu_\omega(X) = 1$ for almost all $\omega$ and $\mu$ being a Young measure w.r.t. $\nu$. The idea is that $\mu_\omega(x)$ is a minimizing density for every single configuration of $\omega$.

*Proof.* We use $\mathcal{P}_{\geq 0}(X \times \Omega)$ to denote the space of non-negative polynomials on $X \times \Omega$. Given measure $\nu$ on $\Omega$ and polynomial function $p : X \times \Omega \to \mathbb{R}$ consider

$$\sup_{\substack{\gamma \in L^1(\Omega,\nu) \\ q \in \mathcal{P}_{\geq 0}(X \times \Omega).}} \int_\Omega \gamma(\omega)\mathrm{d}\nu(\omega)$$

$$\text{s.t} \quad p(x,\omega) - \gamma(\omega) = q(x,\omega)$$

This is equivalent to

$$- \inf_{\substack{\gamma \in L^1(\Omega,\mu) \\ q \in \mathcal{P}_{\geq 0}(X \times \Omega)}} f(\gamma,q) + g(\gamma,q)$$

with

$$f(\gamma,q) = -\int_\Omega \gamma(\omega)\mathrm{d}\nu(\omega)$$

and

$$g(\gamma,q) = -\chi_{\{f-\gamma-q=0\}} = \begin{cases} 0 \text{ if } f - \gamma - q = 0 \\ -\infty \text{ else} \end{cases} ,$$

i.e. $g$ is the characteristic function enforcing non-negativity.

Denote by $h^*$ the Legendre dual, i.e.

$$h^*(y) = \sup_x \langle x, y \rangle - h(x).$$

Then by Rockafellar duality, [47, 24], and noting that signed Borel measures $\mathcal{B}$ are the dual to continuous functions, the dual problem reads

$$\sup_{\Gamma \in L^\infty(\Omega,\mu), \mu \in \mathcal{B}} -f^*(\Gamma,\mu) - g^*(-(\Gamma,\nu))$$

and we would have

$$\sup_{\Gamma \in L^\infty(\Omega,\mu), \mu \in \mathcal{B}} -f^*(\Gamma,\mu) - g^*(-(\Gamma,\mu)) = - \inf_{\substack{\gamma \in L^1(\Omega,\mu) \\ q \in \mathcal{P}_{\geq 0}(X \times \Omega)}} f(\gamma,q) + g(\gamma,q).$$

The Legendre duals of $f$ and $g$ can be explicitly calculated as

$$f^*(\Gamma, \mu) = \begin{cases} 0 \text{ if } \Gamma = -1 \text{ and } \mu \leq 0 \\ \infty \text{ else} \end{cases}$$

and

$$g^*(\Gamma, \mu) = \begin{cases} \int_{\Omega \times X} f(x, \omega) \mathrm{d}\mu(\omega, x) & \text{if } f - \gamma \in \mathcal{P}_{\geq 0}(X \times \Omega) \text{ and } \Gamma(\omega) = \mu_\omega(X) \\ \infty & \text{else} \end{cases}$$

since

$$\begin{aligned} f^*(\Gamma, \mu) &= \sup_{\gamma, q} \left( \int_\Omega \gamma(\omega)\Gamma(\omega)\mathrm{d}\nu(\omega) + \int_{\Omega \times X} q(x, \omega)\mathrm{d}\mu(x, \omega) - f(\gamma, q) \right) \\ &= \sup_{\gamma, q} \int_\Omega \gamma(\omega)(\Gamma(\omega) + 1)\mathrm{d}\nu(\omega) + \int_{\Omega \times X} q(x, \omega)\mathrm{d}\mu(x, \omega) \\ &= \begin{cases} 0 & \text{if } \Gamma = -1 \text{ and } \mu \leq 0 \\ \infty & \text{else} \end{cases} \end{aligned}$$

and

$$\begin{aligned} g^*(\Gamma, \mu) &= \sup_{\gamma, q} \int_\Omega \gamma(\omega)\Gamma(\omega)\mathrm{d}\nu(\omega) + \int_{\Omega \times X} q(x, \omega)\mathrm{d}\mu(\omega, x) + \chi_{\{f-\gamma-q=0\}} \\ &= \begin{cases} \sup_\gamma \int_\Omega \gamma(\omega)\Gamma(\omega)\mathrm{d}\nu(\omega) + \int_{\Omega \times X} (f(x, \omega) - \gamma(\omega))\mathrm{d}\mu(\omega, x) & \text{if } f - \gamma = \in \mathcal{P}_{\geq 0}(X \times \Omega) \\ \infty & \text{else} \end{cases} \\ &= \begin{cases} \sup_\gamma \int_\Omega \gamma(\omega)(\Gamma(\omega) - \mu_\omega(X))\mathrm{d}\nu(\omega) + \int_{\Omega \times X} (f(x, \omega)\mathrm{d}\mu(\omega, x) & \text{if } f - \gamma \in \mathcal{P}_{\geq 0}(X \times \Omega) \\ \infty & \text{else} \end{cases} \\ &= \begin{cases} \int_{\Omega \times X} (f(x, \omega)\mathrm{d}\mu(\omega, x) & \text{if } f - \gamma \in \mathcal{P}_{\geq 0}(X \times \Omega) \text{ and } \Gamma(\omega) = \mu_\omega(X) \\ \infty & \text{else} \end{cases} \end{aligned}$$

Altogether, we get

$$-f^*(\Gamma, \mu) - g^*(-\Gamma, -\mu) = \begin{cases} \int_{\Omega \times X} f(x, \omega)\mathrm{d}\mu(\omega, x) & \text{if } \mu_\omega(X) = 1 \\ \infty & \text{else.} \end{cases}$$

$\square$

### A.5.2 Convergence of S-SOS hierarchy

$c^*(\omega)$ **is at best Lipschitz continuous**    By Proposition 2.1, we argue that $c^*(\omega) = \inf_x f(x, \omega)$ is Lipschitz continuous. One cannot expect much more as the following example shows:

**Example A.4.** Consider $g : \mathbb{R} \times \mathbb{R}^2 \to \mathbb{R}$ defined by

$$g(x, p, q) = (x^2 + px + q)^2.$$

Then we have for every $(p, q) \in \mathbb{R}^2$ that

$$\inf_{x \in \mathbb{R}} g(x, p, q) = \begin{cases} 0 & \text{if } \frac{p^2}{4} \geq q \\ (\frac{p^2}{4} - q)^2 & \text{else.} \end{cases}$$

Therefore, $(p, q) \mapsto \inf_{x \in \mathbb{R}} g(x, p, q)$ is once differentiable but not twice.    $\square$

**Lemma on approximating polynomials**

**Lemma A.5.** *Let $\Omega$ be a compact subset of $\mathbb{R}^n$ and $g : \Omega \to \mathbb{R}$ be Lipschitz continuous. Then there exists a trigonometric polynomial $g_s$ of degree $s$ and a constant $C > 0$ depending only on $\Omega$ and $n$ such that*

$$g \geq g_s$$

*and*

$$\|g - g_s\|_{L^1(\Omega)} \leq \frac{1 + \ln(s)}{s} C \|g\|_{H^1(\Omega)}.$$

*Proof.* By Jackson's inequality for Lipschitz functions [48] we have the existence of a trigonometric polynomial $g'$ of degree $s$ with

$$\|g - g'\|_{L^1(\Omega)} \leq \frac{C'}{s} \|g\|_{H^1(\Omega)}$$

as well as

$$\|g - g'\|_{L^\infty(\Omega)} \leq L_g \frac{\ln(s)}{s}.$$

Then we define $g_s = g' - \|g - g'\|_{L^\infty(\Omega)}$ and hence $g \geq g_s$. Furthermore,

$$\|g - g_s\|_{L^1(\Omega)} \leq \frac{(C' + |\Omega| \ln s)}{s} L_g.$$

Writing $C(\Omega) = \max\{C', |\Omega|\}$ we have the desired form where $|\Omega|$ is the volume of $\Omega$. $\qquad\square$

**Convergence at $\ln s / s$ rate using integral operator methodology**

**Theorem A.1** (Asymptotic convergence of S-SOS). *Let $f : [0,1]^n \times \Omega \to \mathbb{R}$ be a trigonometric polynomial of degree $2r$, $c^*(\omega) = \inf_x f(x, \omega)$ the optimal lower bound as a function of $\omega$, and $\nu$ any probability measure on compact $\Omega \subset \mathbb{R}^d$. Let $s = (s_x, s_\omega, s_c)$, referring separately to the degree of the basis in $x$ terms, the degree of the basis in $\omega$ terms, and the degree of the lower-bounding polynomial $c(\omega)$.*

*Let $c_{2s}^*(\omega)$ be the lower bounding function obtained from the primal S-SOS SDP with $m_s(x, \omega)$ a spanning basis of trigonometric monomials with degree $\leq s_x$ in $x$ terms and of degree $\leq s_\omega$ in $\omega$ terms:*

$$p_{2s}^* = \sup_{c \in \mathcal{P}^{2s_c}(\Omega), W \succcurlyeq 0} \int c(\omega) d\nu(\omega)$$
$$\text{s.t.} \quad f(x, \omega) - c(\omega) = m_s(x, \omega)^T W m_s(x, \omega)$$

*Then there is a constant $C > 0$ depending only on $\Omega, d,$ and $n$ such that for all $s_\omega, s_x \geq \max\{3r, 3s_c\}$ the following holds:*

$$\int_\Omega [c^*(\omega) - c_{2s}^*(\omega)] \, d\nu(\omega) \leq |\Omega| \epsilon(f, s)$$

$$\varepsilon(f, s) \leq \|f - \bar{f}\|_F \left[ 1 - \left(1 - \frac{6r^2}{s_\omega^2}\right)^{-d} \left(1 - \frac{6r^2}{s_x^2}\right)^{-n} \right]$$

$$+ \|c^* - \bar{c}^*\|_F \left[ 1 - \left(1 - \frac{6r^2}{s_\omega^2}\right)^{-d} \right] + C \frac{(1 + \ln(2s_c))}{2s_c}.$$

*where $\bar{f}$ denotes the average value of the function $f$ over $[0,1]^n$, i.e. $\bar{f} = \int_{[0,1]^n} f(x) dx$ and $\|f(x)\|_F = \sum_{\hat{x}} |\hat{f}(\hat{x})|$ denotes the norm of the Fourier coefficients.*

*$\epsilon(f, s)$ bounds the expected error, giving us asymptotic convergence as $s = \min(s_x, s_\omega, s_c) \to \infty$. Note the first two terms give a $O(\frac{1}{s^2})$ convergence rate. However, the overall error will be dominated by the degree of $c(\omega)$ (from the third term) hence our convergence rate is $O(\frac{\ln s}{s})$.*

*Proof of Theorem A.1.* Let $\Omega \subset \mathbb{R}^d$ be compact and $f : R^n \times \Omega \to \mathbb{R}$ be a 1-periodic trigonometric polynomial (t.p.) of degree $\leq 2r$. We then make $\Omega$ isomorphic to $[0,1]^d$ and hereafter consider $\Omega = [0,1]^d$ and $f : [0,1]^n \times [0,1]^d \to \mathbb{R}$. Let $\varepsilon > 0$ and $b = \frac{\varepsilon}{2}$. Let the best lower bound be

$$c^*(\omega) = \inf_{x \in X} f(x, \omega).$$

*Proof outline.* We split the error into two parts. First, we use the fact that there is a lower-bounding t.p. $c_a^*$ of degree $s_c$ such that

$$\|c^* - c_a^*\| \leq C \frac{1 + \ln s_c}{s_c}$$

and

$$c^* \geq c_a^*.$$

This will provide us with a degree-$s_c$ t.p. approximation to the lower bounding function, which in general is only known to be Lipschitz continuous.

Next, we show, that for any $b > 0$ there is a degree-$2s$ SOS t.p. $f_{\text{SOS}}(x, \omega)$ such that

$$f_{\text{SOS}} = f - (c_a^* - b).$$

We write $s = (s_x, s_\omega)$ where $s_x, s_\omega$ denotes the respective max degrees in the variables $x, \omega$. Once we have constructed this, we can compute $f - f_{\text{SOS}} = c_a^* - \varepsilon$ and since we know that $f_{\text{SOS}} \geq 0$ everywhere and $c_a^* - \varepsilon$ is some degree-$s_c$ t.p. we have found a degree-$s_c$ lower-bounding t.p. The construction of this SOS t.p. adds another error term. If we can drive $\varepsilon \to 0$ as $\bar{s} = \min(s_x, s_\omega, s_c) \to \infty$ then we are done.

*Proof continued.* To that end, let $c_a^* : \Omega \to \mathbb{R}$ be the best degree-$s_c$ trigonometric approximation of $c^*$ with respect to $L^1$ such that

$$c^* \geq c_a^*.$$

By [30], we know that $c^*$ is locally Lipschitz continuous with Lipschitz constant $L_{c^*}$ and hence, by Lemma A.5 we get that there is $C(\Omega) > 0$ such that

$$\|c^* - c_a^*\|_{L^1\Omega)} \leq C(\Omega) \frac{1 + \ln s_c}{s_c} L_{c^*}.$$

Next we introduce $c_{2s}^*(\omega)$ which is some degree-$2s$ t.p. After an application of the triangle inequality and Cauchy-Schwarz on the integrated error term $\int_\Omega |c^* - c_{2s}^*| d\omega$ we have

$$\int_\Omega \left| \inf_{x \in X} f(x, \omega) - c_{2s}^*(\omega) \right| d\omega \leq \int_\Omega |c_a^*(\omega) - c_{2s}^*(\omega)| d\omega + |\Omega| \|c^* - c_a^*\|_{L^2(\Omega)}$$

$$\int_\Omega \left| \inf_{x \in X} f(x, \omega) - c_{2s}^*(\omega) \right| d\omega \leq \underbrace{\int_\Omega |c_a^*(\omega) - c_{2s}^*(\omega)| d\omega}_{\text{gap between some SDP solution } c_{2s}^*(\omega) \text{ and t.p. } c_a^*(\omega)} + \underbrace{C(\Omega) \frac{1 + \ln s_c}{s_c} L_{c^*}}_{\text{approx. error of L-contin. fn.}}$$

Now we want to show that for any $\varepsilon > 0$ we can construct a degree-$2s$ SOS trigonometric polynomial $f_{\text{SOS}}(x, \omega)$ such that

$$f_{\text{SOS}} = f - c_a^* + b.$$

with $b = \varepsilon/2$ and $s = (s_x, s_\omega) > r$. We can then set $f - f_{\text{SOS}} = c_a^* - b = c_{2s}^*$ as the degree-$2s$ lower-bounding function. If we can drive $b = \varepsilon/2 \to 0$ as $s, s_c \to \infty$ we are done, as by construction $|c_a^* - c_{2s}^*| = b$.

Observe that by assumption $f - c_a^* + b$ is a t.p. in $(x, \omega)$ where $f$ is degree-$2r$ and $c_a^*$ is degree $s_c \geq 2r$. Denote by $(f - f_*^a + b)_\omega$ its coefficients w.r.t the $\omega$ basis. Note that the coefficients are functions in $x$. Following the integral operator proof methodology in [6], define the integral operator $T$ to be

$$Th(x, \omega) = \int_{X \times \Omega} |q_\omega(\omega - \bar{\omega})|^2 |q_x(x - \bar{x})|^2 h(\bar{x}, \bar{\omega}) d\bar{x} d\bar{\omega},$$

where $q_\omega$ is a trigonometric polynomial in $\omega$ of degree $\leq s_\omega$ and $q_x$ is a trigonometric polynomial in $x$ of degree $\leq s_x$. The intuition is that this integral operator explicitly builds a SOS function of degrees $(s_x, s_\omega)$ out of any non-negative function $h$ by hitting it against the kernels $q_x, q_\omega$.

We want to find a positive function $h : X \times \Omega \to \mathbb{R}$ such that

$$Th = f - c_a^* + b.$$

In frequency space, the Fourier transform turns a convolution into pointwise multiplication so we have:

$$\widehat{Th}(\hat{x}, \hat{\omega}) = \hat{q}_\omega * \hat{q}_\omega(\hat{\omega}) \cdot \hat{q}_x * \hat{q}_x(\hat{x}) \cdot \hat{h}(\hat{x}, \hat{\omega}).$$

In the Fourier domain it is easy to write down the coefficients of $\hat{h}$:

$$\hat{h}(\hat{x}, \hat{\omega}) = \begin{cases} 0 & \text{if } \|\hat{x}, \hat{\omega}\|_\infty > \max\{2r, 2s_c\} \\ \dfrac{\hat{f}(\hat{x}, \hat{\omega}) - \hat{c}_a^*(\hat{\omega}) 1_{\hat{x}=0} + b 1_{\hat{x}=0} 1_{\hat{\omega}=0}}{\hat{q}_\omega * \hat{q}_\omega(\hat{\omega}) \cdot \hat{q}_x * \hat{q}_x(\hat{x})} & \text{otherwise.} \end{cases}$$

Computing $Th - h$ gives:

$$f(x, \omega) - c_a^*(\omega) + b - h(x, \omega)$$

$$= \sum_{\hat{\omega}, \hat{x}} \hat{f}(\hat{x}, \hat{\omega}) \left( 1 - \frac{1}{\hat{q}_\omega * \hat{q}_\omega(\hat{\omega}) \cdot \hat{q}_x * \hat{q}_x(\hat{x})} \right) \exp(2i\pi \hat{\omega}^T \omega) \exp(2i\pi \hat{x}^T x)$$

$$+ \sum_{\hat{\omega}} (b 1_{\hat{\omega}=0} - c_a^*) \left( 1 - \frac{1}{\hat{q}_\omega * \hat{q}_\omega(\hat{\omega})} \right) \exp(2i\pi \hat{\omega}^T \omega)$$

and thus after requiring $\hat{q}_\omega * \hat{q}_\omega(0) = \hat{q}_x * \hat{q}_x(0) = 1$ we have:

$$\max_{x, \omega} |f(x, \omega) - c_a^*(\omega) + b - h(x, \omega)|$$

$$\leq \|f - \bar{f}\|_F \max_{\hat{\omega} \neq 0} \max_{\hat{x} \neq 0} \left| 1 - \frac{1}{\hat{q}_\omega * \hat{q}_\omega(\hat{\omega}) \cdot \hat{q}_x * \hat{q}_x(\hat{x})} \right|$$

$$+ \max_{\hat{\omega} \neq 0} \|c_a^* - \bar{c}_a^*\|_F \left| 1 - \frac{1}{\hat{q}_\omega * \hat{q}_\omega(\hat{\omega})} \right|.$$

As a reminder, because $c^* \geq c_a^*$ everywhere we have $f - c_a \geq f - c^* \geq 0$ or $f - c_a^* + b > 0$, since $b = \varepsilon/2 > 0$. Since $Th = f - c_a^* + b > 0$ and it is a SOS, we need to guarantee $h > 0$.

If $\max_{x, \omega} |f(x, \omega) - f_*^a(\omega) + b - h(x, \omega)| \leq b$ then

$$\max_{x, \omega} |Th - h| < b.$$

Since $Th \geq b$ and $b > 0$ we have

$$h = Th + h - Th \geq Th - \|h - Th\|_\infty \geq b - b \geq 0$$

and hence $h > 0$ if we ensure $\max_{x, \omega} |Th - h| \leq b$.

Now let us show that

$$\max_{x, \omega} |f(x, \omega) - c_a^*(\omega) + b - h(x, \omega)| \leq b$$

can be ensured if $s = (s_x, s_\omega)$ is large enough.

Using the same kernel and bounds as in [6], we choose for $z \in \{x, \omega\}$ the triangular kernel such that

$$\hat{q}_z(\hat{z}) = \left( 1 - \frac{6r^2}{z^2} \right)_+^d \prod_{i=1}^d \left( 1 - \frac{|\hat{z}_i|}{s_{x, \omega}} \right)_+.$$

Note that $(x)_+ = \max(x, 0)$. Then we have

$$\max_x |f(x, \omega) - c_a^*(\omega) + b - h(x, \omega)|$$

$$\leq \|f - \bar{f}\|_F \max_{\hat{\omega}, \hat{x}} \left| 1 - \frac{1}{\hat{q}_\omega * \hat{q}_\omega(\hat{\omega}) \cdot \hat{q}_x * \hat{q}_x(\hat{x})} \right| + \|c_a^* - \bar{c}_a^*\|_F \max_{\hat{\omega}} \left| 1 - \frac{1}{\hat{q}_\omega * \hat{q}_\omega(\hat{\omega})} \right|$$

$$\leq \|f - \bar{f}\|_F \left| 1 - \left( 1 - \frac{6r^2}{s_\omega^2} \right)^{-d} \left( 1 - \frac{6r^2}{s_x^2} \right)^{-n} \right| + \|c_a^* - \bar{c}_a^*\|_F \left| 1 - \left( 1 - \frac{6^2}{s_\omega^2} \right)^{-d} \right|$$

Therefore, by choosing $s_\omega$ and $s_x$ large enough such that

$$\|f - \bar{f}\|_F \left|1 - \left(1 - \frac{6r^2}{s_\omega^2}\right)^{-d}\left(1 - \frac{6r^2}{s_x^2}\right)^{-n}\right| + \|c_a^* - \bar{c}_a^*\|_F\left|1 - \left(1 - \frac{6^2}{s_\omega^2}\right)^{-d}\right| \le b = \frac{\varepsilon}{2}$$

we have

$$h \ge 0$$

and thus $Th$ is SOS. By design we have

$$c_a^* - c_{2s}^* \le b$$

and thus

$$\int_\Omega |c_a^* - c_{2s}^*|\mathrm{d}\omega \le \frac{\varepsilon}{2}.$$

Recalling

$$\int_\Omega \left|\inf_{x \in X} f(x,\omega) - c_{2s}^*(\omega)\right|\mathrm{d}\omega \le \underbrace{\int_\Omega |c_a^*(\omega) - c_{2s}^*(\omega)|\mathrm{d}\omega}_{\text{gap between some SDP solution } c_{2s}^*(\omega) \text{ and t.p. } c_a^*(\omega)} + \underbrace{C(\Omega)\frac{1 + \ln s_c}{s_c}L_{c^*}}_{\text{approx. error of L-contin. fn.}}$$

we can additionally choose $s_c$ large enough to guarantee

$$C(\Omega)\frac{1 + \ln s_c}{s_c}L_{c^*} \le \frac{\varepsilon}{2}$$

and then we are done.

Setting $s_x, s_\omega, s_c = s$ and sending $s \to \infty$ we have asymptotic behavior of the final error expression:

$$\boxed{\int_\Omega \left|\inf_{x \in X} f(x,\omega) - c_{2s}^*(\omega)\right|\mathrm{d}\omega \le C_1\frac{1}{s^2} + C_2\frac{1}{s} + C_3\frac{\ln s}{s} = \mathcal{O}\left(\frac{\ln s}{s}\right)}$$

with the constants $C_1, C_2, C_3$ depending on $r, n, d, \|f - \bar{f}\|_F, \|c_a - \bar{c}_a^*\|_F, \Omega$ and $L_{c^*}$. $\qquad\square$

**Convergence at $1/s$ rate using piecewise-constant approximation to $c^*(\omega)$**  Prior work [6] achieves $1/s^2$ convergence for the regular SOS hierarchy without further assumptions. In our work, we could only achieve $\ln s/s$ for S-SOS due to the need to first approximate the tightest lower-bounding function $c^*(\omega)$ with a polynomial approximation, which converges at a slower rate.

To accelerate the convergence rate, we want to control the regularity of $c^*(\omega)$. We can achieve $1/s$ by approximating the $c^*(\omega)$ pointwise instead of using a smooth parameterized polynomial. By constructing a domain decomposition of $\Omega$ and finding a SOS approximation in $x$ for each domain, we can stitch these together to build a piecewise-constant approximation to the lower-bounding function $c^*$. This lets us leverage the $1/s^2$ rate for the standard SOS hierarchy, however note that we lose the guarantee that the lower-bounding function $c_s^*(\omega)$ is a true lower bound everywhere. Another disadvantage of this approach is that the computational complexity scales exponentially in $d$ as we are required to solve for the SOS lower bound with degree $s$ pointwise at $O(s_p^d)$ points.

For $\Omega = [0,1]^d \subset \mathbb{R}^d$ we achieve the following:

**Proposition A.6.** *Let $\Omega = [0,1]^d$ and $f : [0,1]^n \to \mathbb{R}$ be a trigonometric polynomial of degree $2r$. Let $\{\omega_i\}$ be grid points where $s_p$ is the number of equally-spaced points along each dimension for a total of $(s_p)^d$ points in $\Omega$. Denote by $c_s^*(\omega_i)$ the best SOS approximation of degree $s$ of $x \mapsto f(x,\omega_i)$ at each grid point $\omega_i$ and define*

$$c_s^*(\omega) = \sum_{i=1}^{s_p^d} c_s^*(\omega_i)\mathbb{1}_{\omega \in \mathcal{B}_\infty(\omega_i,\epsilon)}$$

*where $\mathcal{B}_\infty(\omega_i,\epsilon)$ denotes the $L_\infty$ ball of radius $\epsilon$ centered at $\omega_i$ with $\epsilon = (2s_p)^{-1}$ so that the balls are non-overlapping. Then we have for some constant $C'$ depending only on $\max_{\omega_i} \|f(\omega_i,\cdot) - \bar{f}(\omega_i,\cdot)\|_F, r, n, d, s_p$:*

$$\int_\Omega [c^*(\omega) - c_s^*(\omega)]\mathrm{d}\omega \le \max_{\omega_i}\|f(\omega_i,\cdot) - \bar{f}(\omega_i,\cdot)\|_F\left[1 - \left(1 - \frac{6r^2}{s^2}\right)^{-n}\right] + \frac{C\sqrt{d}}{s_p} \le \frac{C'}{s}.$$

*Proof of Proposition A.6.* Let $c_a^*(\omega) : [0,1]^d \to \mathbb{R}$ be the best piecewise-constant approximation of the true lower-bounding function $c^*(\omega) = \inf_x f(x,\omega)$ on equidistant grid-points, i.e. $s_p$ evenly-spaced points in each dimension for a total of $s_p^d$ points.

Now also define a piecewise-constant approximation using the best pointwise SOS lower bounds:

$$c_s^*(\omega) = \sum_{i=1}^{s_p^d} c_s^*(\omega_i) 1_{\omega \in \mathcal{B}_\infty(\omega_i,\epsilon)}$$

where $c_s^*(\omega_i)$ is the best lower bound (resulting from regular SOS) of degree $s$ of $x \mapsto f(x,\omega_i)$ and $\mathcal{B}_\infty(\omega_i,\epsilon)$ denotes the $L_\infty$ ball of radius $\epsilon$ centered at $\omega_i$ with $\epsilon = (2s_p)^{-1}$ so that the balls are non-overlapping.

By [6] we have that $c_a^*(\omega_i) - c_s^*(\omega_i)$ can be bounded by

$$\max_{\omega_i} \|f(\omega_i,\cdot) - \bar{f}(\omega_i,\cdot)\|_F \left(1 - \left(1 - \frac{6r^2}{s^2}\right)^{-n}\right).$$

Then we have:

$$\int_\Omega [c^*(\omega) - c_s^*(\omega)]d\omega \leq \sum_{i=1}^{s_p^d} |c_a^*(\omega_i) - c_s^*(\omega_i)|(2\epsilon)^d + \|c^* - c_a^*\|_{L^1(\Omega)}.$$

Using the same bound we get for the first term from the proof of Theorem A.1, we can reduce the first term to a $O(1/s^2)$ dependence and we use the theorem on the $L^1$ convergence of piecewise-constant approximation to 1-periodic trigonometric polynomials from [48] for the second:

$$\int_\Omega [c^*(\omega)s - c_s^*(\omega)]d\omega \leq \max_{\omega_i} \|f(\cdot,\omega_i) - \bar{f}(\cdot,\omega_i)\|_F \left(1 - \left(1 - \frac{6r^2}{s^2}\right)^{-n}\right)|\Omega| + \frac{C\sqrt{d}}{s_p}$$

It is straightforward to extend this to arbitrary compact $\Omega \subset \mathbb{R}^d$, as one can construct a smooth homeomorphism between $\Omega$ and $[0,1]^d$ and therefore we can achieve the same convergence rates on general compact $\Omega$ as we can for $[0,1]^d$. □

## A.6 S-SOS for a simple quadratic potential

We provide a simple application of S-SOS to a simple quadratic potential that admits a closed-form solution so as to demonstrate its usage and limitations.

### A.6.1 Analytic solution for the lower bounding function $c^*(\omega)$ with $\omega \sim$ **Uniform**$(-1,1)$

Let $x \in \mathbb{R}$ and $\omega \sim \text{Uniform}(-1,1)$. Suppose that we have

$$f(x,\omega) = (x - \omega)^2 + (\omega x)^2$$

In this case we may explicitly evaluate the exact minimum function $c^*(\omega) = \inf_x f(x;\omega)$. Note that

$$f(x;\omega) = x^2 - 2\omega x + \omega^2 + \omega^2 x^2$$

Explicitly evaluating the zeros of the first derivative we have

$$\partial_x f(x;\omega) = 2x^* - 2\omega + 2\omega^2 x^* = 0$$
$$x^*(1 + \omega^2) = \omega$$
$$x^* = \frac{\omega}{1 + \omega^2}$$

and, thus,

$$c^*(\omega) = \inf_x f(x;\omega) = \frac{\omega^4}{1 + \omega^2}.$$

Note that despite $f(x,\omega)$ being a simple degree-2 SOS polynomial, the tightest lower-bound $c^*(\omega) = \inf_x f(x,\omega)$ is explicitly not polynomial. However, it is algebraic, as it is defined implicitly as the root of the polynomial equation

$$c^*(\omega)(1 + \omega^2) - \omega^4 = 0$$

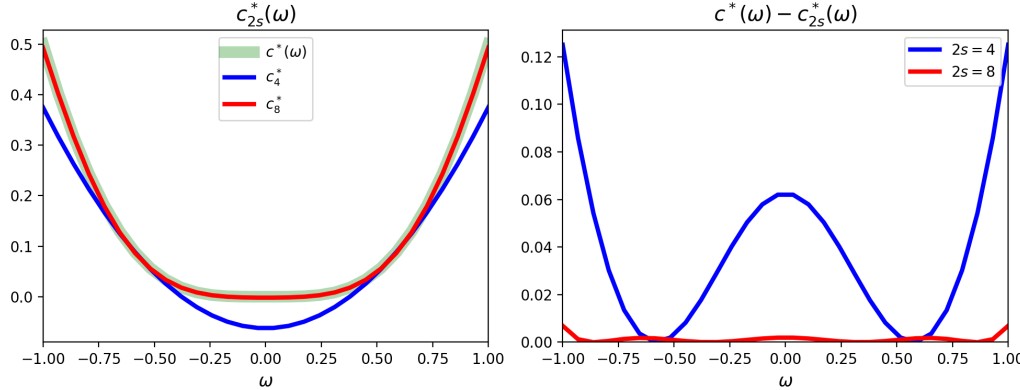

Figure 2: Lower bound functions for basis function degree $d = 2, 4$ (left) and the optimality gap to the true lower bound $c^*(\omega) - c^*_{2s}(\omega)$ (right)

### A.6.2 Degree-$2s$ S-SOS to find a polynomial lower-bounding function $c^*_{2s}(\omega)$

Observe that the tightest lower-bounding function $c^*(\omega)$ is not polynomial even in this simple setting. However, we can relax the problem to trying to find $c_{2s} \in \mathcal{P}^{2s}(\Omega)$ to obtain a weaker bound with $\inf_x f(x, \omega) = c^*(\omega) \geq c_{2s}(\omega)$.

We now proceed with formulating and solving the degree-$2s$ primal S-SOS SDP (Equation (2)). We assume that $c_{2s}(\omega)$ is parameterized by a polynomial of degree $\leq 2s$ in $\omega$. Observe that this class of functions is not large enough to contain the true function $c^*(\omega)$.

We choose $s \in \{2, 4\}$ and use the standard monomial basis in $x, \omega$, we have the feature maps $m_2(x, \omega) : \mathbb{R}^2 \to \mathbb{R}^6$ and $m_4(x, \omega) : \mathbb{R}^2 \to \mathbb{R}^{15}$, since there are $\binom{n+s}{s}$ unique monomials of up to degree-$s$ in $n$ variables. These assumptions together enable us to explicitly write a SOS SDP in terms of coefficient matching. Note that we must assume some noise distribution $\nu(\omega)$. For this section, we present results assuming $\omega \sim \text{Uniform}(-1, 1)$. We solve the resulting SDP in CVXPY using Legendre quadrature with $k = 5$ zeroes on $[-1, 1]$ to evaluate the objective $\int c(\omega) d\nu(\omega)$. In fact, $k$ sample points suffice to exactly integrate polynomials of degree $\leq 2k - 1$.

We solve the SDP for two different levels of the hierarchy, $s = 2$ and $s = 4$ (producing lower-bound polynomials of degree 4 and 8 respectively), and plot the lower bound functions $c_{2s}(\omega)$ vs the true lower bound $c^*(\omega) = \omega^4/(1 + \omega^2)$ as well as the optimality gap to the true lower bound in Fig.2.

### A.6.3 Convergence of lower bound as degree $s$ increases

To solve the S-SOS SDP in practice, we must choose a maximum degree $2s$ for the SOS function $m_2(x, \omega)^T W m_2(x, \omega)$ and the lower-bounding function $c(\omega)$, which are both restricted to be polynomials. Indeed, a larger $s$ not only increases the dimension of our basis function $m_s(x, \omega)$ but also the complexity of the resulting SDP. We would expect that $d^*_{2s} \to d^*$ as $s \to \infty$, i.e. the optimal value of the degree-$2s$ S-SOS SDP (Equation (4)) converges to that of the "minimizing distribution" optimization problem (Equation (3)).

In particular, note that in the standard SOS hierarchy we typically find finite convergence (exact agreement at some degree $2s^* < \infty$). However, in S-SOS, we thus far have only a guarantee of asymptotic convergence, as each finite-degree S-SOS SDP solves for a polynomial approximation to the optimal lower bound $c^*(\omega) = \inf_{x \in X} f(x, \omega)$. In Figure 1, we illustrate the primal S-SOS SDP objective values

$$p^*_{2s} = \sup_{c \in \mathcal{P}^{2s}(\Omega)} \int c(\omega) d\nu(\omega) \quad \text{with} \quad f(x, \omega) - c(\omega) \in \mathcal{P}^{2s}_{\text{SOS}}(X \times \Omega)$$

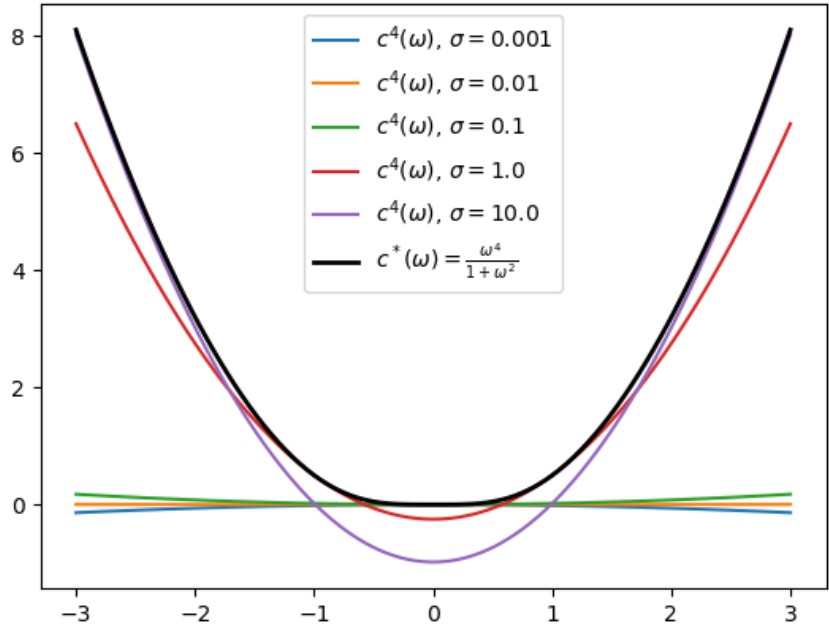

Figure 3: Different lower-bounding functions for degree-4 S-SOS done on the simple quadratic potential $f(x, \omega) = (x - \omega)^2 + (\omega x)^2$. The true lower-bounding function $c^*(\omega)$ is plotted in black.

for a given level of the hierarchy (a chosen degree $s$ for the basis $m_s(x, \omega)$) and their convergence towards the optimal objective value

$$\int c^*(\omega) d\nu(\omega) = \frac{\pi}{4} - \frac{2}{3} \approx 0.1187$$

for the simple quadratic potential, assuming $\nu(\omega) = \frac{1}{2}$ with $\omega \sim \text{Uniform}(-1, 1)$. We note that in the log-linear plot (right) we have a "hinge"-type curve, with a linear decay (in logspace) and then flattening completely. This suggests perhaps that in realistic scenarios the degree needed to achieve a close approximation is very low, lower than suggested by our bounds. The flattening that occurs here is likely due to the numerical tolerance used in our solver (CVXPY/MOSEK), as increasing the tolerance also increases the asymptotic gap and decreases the degree at which the gap flattens out.

### A.6.4 Effect of different noise distributions

In the previous two sections, we assumed that $\omega \sim \text{Uniform}(-1, 1)$. This enabled us to solve the primal exactly using Legendre quadrature of polynomials. Note that in Figure 1 we see that the lower-bounding $c_2^*(\omega), c_4^*(\omega)$ for $\omega \sim \text{Uniform}(-1, 1)$ is a smooth polynomial that has curvature (i.e. sign matching that of the true minimum). This is actually not guaranteed, as we will see shortly.

In Figure 3, we present the lower-bounding functions $c_4^*(\omega)$ achieved by degree-4 S-SOS by solving the dual for $\omega \sim \text{Normal}(0, \sigma^2)$ for varying widths $\sigma$. We can see that for small $\sigma \ll 1$, the primal solution only cares about the lower-bound accuracy within a small region of $\omega = 0$, and the lower-bounding curve fails to "generalize" effectively outside the region of consideration.

### A.7 S-SOS for sensor network localization

### A.7.1 SDP formulation

Recall the form of $f(x, \omega)$:

$$f(x, \omega; X, A, r) = \underbrace{\sum_{d_{ij} \in \mathcal{D}_{ss}(r)} (\|x_i - x_j\|_2^2 - d_{ij}(\omega)^2)^2}_{\text{sensor-sensor interactions}} + \underbrace{\sum_{d_{ik} \in \mathcal{D}_{sa}(r)} (\|x_i - a_k\|_2^2 - d_{ik}(\omega)^2)^2}_{\text{sensor-anchor interactions}}$$

Note that the function $f(x, \omega)$ is exactly a degree-4 SOS polynomial, so it suffices to choose the degree-2 monomial basis containing $a = \binom{N\ell+d+2}{2}$ elements as $m_2(x, \omega) : \mathbb{R}^{N\ell+d} \to \mathbb{R}^a$. That is, we have $N$ sensor positions in $\ell$ spatial dimensions and $d$ parameters for a total of $N\ell + d$ variables.

Let the moment matrix be $M \in \mathbb{R}^{a \times a}$ with elements defined as

$$M_{i,j} := \int m_2^{(i)}(x, \omega) m_2^{(j)}(x, \omega) d\mu(x, \omega)$$

for $i, j \in \{1, \ldots, a\}$, which fully specifies the minimizing distribution $\mu(x, \omega)$ as in Equation (4).

Our SDP is then of the form

$$d_4^* = \inf_y \sum_\alpha f_\alpha y_\alpha$$
$$\text{s.t. } M(y) \succcurlyeq 0$$
$$y_\alpha = m_\alpha \ \forall \ (\alpha, m_\alpha) \in \mathcal{M}_\nu$$
$$y_\alpha = y_\alpha^* \ \forall \ (\alpha, y_\alpha^*) \in \mathcal{H}$$

where $y_\alpha = m_\alpha$ corresponds to the moment-matching constraints of Equation (4) and $y_\alpha = y_\alpha^*$ correspond to any possible hard equality constraints required to set the exact position (and uncertainty) of a sensor $\mathbb{E}[x_i] = x_i^*, \mathbb{E}[x_i^2] - \mathbb{E}[x_i]^2 = 0$ for all $\omega$. $\mathcal{M}_\nu$ represents the $\binom{d+2s}{2s}$ moment-matching constraints necessary for all moments w.r.t. $\omega$ and $\mathcal{H}$ represents the $2\ell n$ constraints needed to set the exact positions of $n$ known sensor positions in $\mathbb{R}^\ell$ (i.e. 1 constraint per sensor and dimension, 2 each for mean and variance).

### A.7.2 Noise types

In this paper we focus on the linear uniform noise case, as it is a more accurate reflection of measurement noise in true SNL problems. Special robust estimation approaches may be needed to properly handle the outlier noise case.

- **Linear uniform noise**: for a subset of edges we write $d_{ij,k}(\omega) = d_{ij}^* + \epsilon \omega_k$, $\omega_k \sim$ Uniform$(-1, 1)$, and $\epsilon \geq 0$ some noise scale we set. The same random variate $\omega_k$ may perturb any number of edges. Otherwise the observed distances are the true distances.

- **Outlier uniform noise**: for a subset of edges we ignore any information in the actual measurement $d_{ij,k} = \omega_k$, $\omega_k \sim$ Uniform$(0, 2\sqrt{\ell})$ where $\ell$ is the physical dimension of the problem, i.e. $x_i \in \mathbb{R}^\ell$.

### A.7.3 Algorithms: S-SOS and MCPO

Here we explicitly formulate MCPO and S-SOS as algorithms. Let $X = \mathbb{R}^n, \Omega = \mathbb{R}^d$ and use the standard monomial basis. We write $z = [x_1, \ldots, x_n, \omega_1, \ldots, \omega_d]$. Our objective is to approximate $c^*(\omega) = \inf_x f(x, \omega)$ for all $\omega$, with a view towards maximizing $\int c^*(\omega) d\nu(\omega)$ for $\omega$ sampled from some probability density $\nu(\omega)$.

MCPO (Algorithm 1) simply samples $\omega_t$ and finds a set of tuples $(x^*(\omega_t), \omega_t)$ where the optimal minimizer $(x^*(\omega_t, \omega_t)$ is computed using a local optimization scheme (we use BFGS).

S-SOS (Algorithm 2) via solving the dual (Equation (4)) is also detailed below.

### A.7.4 Cluster basis hierarchy

Recall from Section 2.2.2 that we defined the cluster basis hierarchy using body order $b$ and maximum degree per variable $t$. In this section, we review the additional modifications needed to scale S-SOS for SNL.

In SNL, $f(x, \omega)$ is by design a degree $s = 4$ polynomial in $z = [x, \omega]$, with interactions of body order $b = 2$ (due to the $(x_i, x_j)$ interactions) and maximum individual variable degree $t = 4$. Written this way, we want to only consider monomial terms $[x, \omega]^\alpha$ with $||\alpha||_1 \leq s$, $||\alpha||_\infty \leq 4$, and $||\alpha||_0 \leq 2$.

To sparsify our problem, we start with some $k$-clustering ($k$ clusters, mutually-exclusive) of the sensor set $\mathcal{C} = \{C_1, \ldots, C_k\}$. This clustering can be considered as leveraging some kind of "coarse"

---

**Algorithm 1** Monte Carlo Point Optimization (MCPO)

---

1: **Input:** Function $f(x; \omega)$, sampler for distribution $\nu(\omega)$, number of samples $T$

2: **Output:** Approximate integral $\hat{I}$, empirical distribution $p_{\mathcal{D}}(x)$, empirical mean $\mu$, empirical covariance $\Sigma$

3: **for** $t = 1$ to $T$ **do**

4:     Sample $\omega_t \sim \nu(\omega)$

5:     Find minimizer $x_t = \min_x f(x; \omega_t)$ using BFGS

6: **end for**

7: Estimate integral $\hat{I} \approx \frac{1}{T} \sum_{t=1}^{T} f(x_t; \omega_t)$

8: Construct empirical distribution

$$p_{\mathcal{D}}(x) = \frac{1}{T} \sum_{t=1}^{T} \delta(x - x_t)$$

9: Calculate empirical mean $\hat{\mu} = \frac{1}{T} \sum_{t=1}^{T} x_t$ and covariance $\hat{\Sigma} = \frac{1}{T-1} \sum_{t=1}^{T} (x_t - \hat{\mu})(x_t - \hat{\mu})^T$.

---

**Algorithm 2** Stochastic Sum-of-squares (S-SOS), Dual formulation

---

1: **Input:** Maximum basis function degree $s \in \mathbb{Z}_{>0}$, complete basis function $m_s(x, \omega) : \mathbb{R}^{n+d} \to \mathbb{R}^{\binom{n+d+s}{s}}$, function $f(x; \omega) : \mathbb{R}^{n+d} \to \mathbb{R}$ represented as a dictionary mapping multi-index $\alpha \in \mathbb{Z}_{\geq 0}^{n+d} \to$ coefficient $f_\alpha$, probability density function for $\nu(\omega)$ with known moments $\int \omega^\alpha d\nu(\omega) < \infty \ \forall \ ||\alpha||_1 \leq 2s$, any hard equality constraints where we want to set $x_k = x_k^*$ for some $k \in \mathcal{K}$.

2: Let $i1, i2, i4$ be the lexicographically-ordered arrays

$$\mathbb{Z}_{\geq 0}^{(n+d+1) \times (n+d)}, \mathbb{Z}_{\geq 0}^{\binom{n+d+s}{s} \times (n+d)}, \mathbb{Z}_{\geq 0}^{\binom{n+d+2s}{2s} \times (n+d)}$$

which correspond to the arrays of multi-indices for all degree-1, degree-$s$, and degree-$2s$ monomials in the variables $z$.

3: Create $M \in \mathbb{R}^{\binom{n+d+s}{s} \times \binom{n+d+s}{s}}$ as a matrix of variables to be estimated.

4: Create $y \in \mathbb{R}^{\binom{n+d+2s}{2s}}$ as a vector of variables to be estimated, corresponding to the vector of independent moments that fully specifies $M$.

5: Add $M \succeq 0$ constraint.

6: **for** $i$ in length($i2$) **do**

7:     **for** $j$ in length($i2$) **do**                    ▷ Require $M$ to be formed from the elements of $y$.

8:         Compute $\alpha_{ij} = i2[i] + i2[j]$ as the multi-index corresponding to the sum of the multi-indices $i2[i], i2[j]$.

9:         Add constraint $M_{i,j} = y_{\alpha_{ij}}$.

10:     **end for**

11: **end for**

12: **for** each row $\alpha$ in $i4$ **do**            ▷ Require $y_\alpha$ moments to equal the known moments of $\omega^\alpha$.

13:     **if** $\sum_{i=1}^{n} \alpha_i = 0$ **then**

14:         Add constraint $y_\alpha = \int z^\alpha d\nu(\omega) = \int \omega^{\alpha[-d:]} d\nu(\omega)$.

15:     **end if**

16: **end for**

17: **for** $k$ in $\mathcal{K}$ **do**                        ▷ Handle any hard equality constraints in our variables $x$.

18:     Form multi-index $\alpha_1 \in \mathbb{Z}_{\geq 0}^{n+d}$ where the entry for $x_k$ is set to 1 and everything else is zero.

19:     Form multi-index $\alpha_2 \in \mathbb{Z}_{\geq 0}^{n+d}$ where the entry for $x_k^2$ is set to 1 and everything else is zero.

20:     Add constraint $y_{\alpha_1} = x_k^*$.                                    ▷ $\mathbb{E}[x_k] = x_k^*$.

21:     Add constraint $y_{\alpha_2} = (x_k^*)^2$.                    ▷ $\mathrm{Var}[x_k] = \mathbb{E}[x_k^2] - \mathbb{E}[x_k]^2 = 0$.

22: **end for**

23: Form the objective to be minimized: $F = \int f(x, \omega) d\mu(x, \omega) = \sum_{\alpha \in i4} f_\alpha y_\alpha$.

24: Solve SDP where we compute $\inf F$ subject to above constraints.

25: **Output:** If the problem is feasible (i.e. there exists a degree-$2s$ decomposition of $f$ into $f_{\mathrm{SOS}}$ and $c_{2s}^*(\omega)$), return moment matrix $M \in \mathbb{R}^{\binom{n+d+s}{s} \times \binom{n+d+s}{s}}$, dual objective value $d_{2s}^*$. Otherwise, terminate and return failed/infeasible SDP solve.

---

information about which sensors are close to each other. For example, just looking at the polynomial $f(x, \omega)$ enables us to see which sensors $(i, j)$ must be interacting.

Assume that there is some *a priori* clustering given to us. We denote $x^{(i)}$ as the subset of the variables restricted to the cluster $C_i$, i.e. $x^{(i)} = \{x_j : j \in C_i\}$. Moreover, let $G = (V, E)$ be a graph where the vertices $V = \{1, \ldots, k\}$ correspond to the $k$ clusters and the edges $E = \{(i, j) : i, j \in V\}$ correspond to known cluster-cluster interactions.

The SOS part of the function $f(x)$ may then be approximated as the sum of dense intra-cluster interactions and sparse inter-cluster interactions, where the cluster-cluster interactions are given exactly by edges in the graph $G$:

$$m_s(x)^T W m_s(x) \approx \sum_{i \in V} m_s(x^{(i)})^T W^{(i)} m_s(x^{(i)}) + \sum_{(i,j) \in E} m_s(x^{(i)})^T W^{(i,j)} m_s(x^{(j)})$$

where $W^{(k)}$ are symmetric PSD matrices and $W^{(i,j)}$ are rectangular matrices where we require $W^{(i,j)} = (W^{(j,i)})^T$. $m_s(x)$ for $x \in \mathbb{R}^n$ here behaves as before and denotes the basis function generated by all $\binom{n+s}{s}$ combinations of monomials with degree $\leq s$. Notice that this is a strict reduction from the standard Lasserre hierarchy at the same degree $s$, since in general the standard basis $m_s(x)$ on the full variable set will contain terms that mix variables from two different clusters that may not have an edge connecting them.

Efficiency gains in the SDP solve occur when we constrain certain of the off-diagonal $W^{(i,j)}$ blocks to be zero, i.e. the graph $G$ is sparse in cluster-cluster interactions. As we can see from the block decomposition written above, this resembles block sparsity on the matrix $W$. We may interpret the above scheme as having a hierarchical structure out to depth 2, where we have dense interactions at the lowest level and sparse interactions aggregating them. In full generality, the resulting hierarchical sparsity in $W$ may be interpreted as generating a chordal $W$, which is known to admit certain speed-ups in SDP solvers [26].

When attempting to solve an SNL problem in the cluster basis instead of the full basis, we need to throw away terms in the potential $f(x, \omega)$ that correspond to cross-terms that are "ignored" by the particular cluster basis we chose. The resulting polynomial $\bar{f}(x, \omega)$ has fewer terms and produces a cluster basis SDP that is easier to solve, but generally less accurate due to the sparser connectivity.

In particular, for the rows in Table 1 that have $N_C > 1$, we do a $N_C$-means clustering of the ground-truth sensor positions and use those sensor labels to create our partitioning of the sensors. We connect every cluster using plus-one $c_i, c_{i+1}$ (including the wrap-around one) connections, so that the cluster-cluster connectivity graph has $N_C$ edges. We then use this information to throw out observed distances from the set $\mathcal{D}_{ss}$ and from the full basis function $m_2(x, \omega)$. See our code for complete details.

### A.7.5 Hard equality constraints

The sensor-anchor terms in Equation (7) are added to make the problem easier, because by adding them now each sensor no longer needs to rely only on a local neighborhood of sensors to localize itself, but can also use its position relative to some known anchor. When we remove them entirely, we need to incorporate hard equality constraints between certain sensors and known "anchor" positions. This fixes certain known sensors but lets every other sensor be unrooted, defined only relative to other sensors (and potentially an anchor if it is within the sensing radius).

To deal with the equality constraints where we set the exact position of a sensor $x_i = x_i^*$, we solve the dual Equation (4) and implement them as equality constraints on the moment matrix, i.e. for the basis element $m_2(x, \omega)_i = x_i$ we may set $\mathbb{E}[x_i] - x_i^* = M_{0,i} - x_i^* = 0$. Note that we also need to set $\text{Var}(x_i) = 0$ so for $m_2(x, \omega)_j = x_i^2$ we add the equality constraint $\text{Var}(x_i) = \mathbb{E}[x_i^2] - \mathbb{E}[x_i]^2 = M_{0,j} - M_{0,i}^2 = 0$.

### A.7.6 Solution extraction

Once the dual SDP has been solved, we extract the moment matrix $M$ and can easily recover the point and uncertainty estimates for the sensor positions $\mathbb{E}[x], \text{Var}[x]$ by inspecting the appropriate entries $M_{0,i}$ corresponding to $m_2(x, \omega)_i = x_i$ and $M_{0,j}$ corresponding to $m_2(x, \omega)_j = x_i^2$.

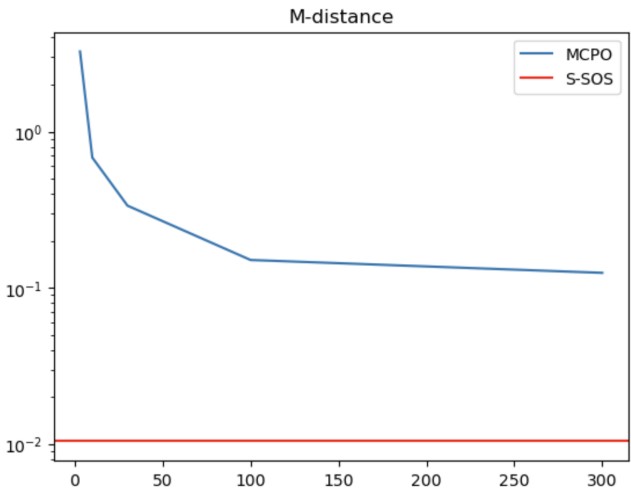

Figure 4: Comparison of the performance of MCPO and S-SOS (degree-4) for sensor recovery accuracy in 1D SNL with varying number of samples $T$ used in the estimate of empirical $\hat{\mu}, \hat{\Sigma}$. M-distance is $\delta_M$, our metric for sensor recovery accuracy per Equation (8). The problem type here is a $N = 5$ sensor, $\ell = 1$ spatial dimension, $|\Omega| = 2$ noise variables, $\epsilon = 0.1$ noise scale, $r = 3$ sensing radius problem. The full basis is used here for the S-SOS SDP.

### A.7.7 Impact of using MCPO with varying numbers of samples $T$

In Figure 4 we can see how $\delta_M$ varies as we scale the number of samples $T$ used in the MCPO estimate of the empirical mean/covariance of the recovered solutions. In this particular example, the runtime of the S-SOS estimate was 0.3 seconds, comparing to 30 seconds for the $T = 300$ MCPO point. Despite taking ~100x longer, the MCPO solution recovery still dramatically underperforms S-SOS in $\delta_M$. This reflects the poor performance of local optimization methods vs. a global optimization method (when it is available).

### A.7.8 Scalability

The largest 2D SNL experiment we could run had $N = 15$ sensors, $N_C = 9$ clusters, and $d = 9$ noise parameters. This generated $N\ell + d = 39$ variables and 820 basis elements in the naive $m_2(x, \omega)$ construction, which was reduced to 317 after our application of the cluster basis, giving us $W, M \in \mathbb{R}^{317 \times 317}$. A single solve in CVXPY (MOSEK) took 30 minutes on our workstation (2x Intel Xeon 6130 Gold and 256GB of RAM). We attempted a run with $N = 20$ sensors and $N_C = 9$ clusters and $d = 9$ noise parameters, but the process failed due to OOM constraints. Thus, we report the largest experiment that succeeded.

### A.8 Limitations

As discussed in the main text, our work provides both a theoretical convergence rate as well as numerical understanding of the stochastic sum-of-squares (S-SOS). Notably, our convergence rate is bottlenecked on the approximation quality of $c^*(\omega)$, only achieving $\ln s/s$ where regular SOS results achieve $1/s^2$. In practice, we see much faster convergence, which may be a clue that other more restrictive assumptions or more powerful methods may enable one to find better convergence results.

In addition, our numerical work focuses on the sensor network localization problem. Due to the time complexity of SDPs, the maximum number of sensors we could solve in any one instance is limited to $N = 15$. This seems much smaller than the state-of-the-art, i.e. [49] where the authors propose an algebraic reduction of the noiseless SNL problem so that they can analytically obtain the range of the PSD matrix. This simplifies the SDP dramatically, however in the noisy setting (where observed sensor-sensor distances are perturbed with observation noise) this approach is unusable, requiring significant modification. his reduction is used to solve SNL problems of 10k-100k sensors, but they develop a highly specialized algorithm that does not use any SDP solvers.

Note that noiseless SNL simplifies the problem. In the noiseless setting, localizing even a small number of sensors near an anchor will propagate the correctly localized positions to sensors nearby. As such, if one can find small groups of sensors that can be well-oriented with respect to each other, one can pursue localization of these groups in parallel and then orient them globally at the very end. In the noisy setting, any error in localization can easily propagate to the whole instance, dramatically reducing the size of problems that can be effectively solved [50, 51].

Thus, we observe that uncertainty makes the SNL problem considerably more difficult than the noiseless setting. S-SOS handles this naturally while also solving the problem for its global optimum. We get a solution for the sensor positions for every possible configuration of noise via the probability distribution $\mu(x, \omega)$. This additional complexity places further limits on the size of the problems that can be solved with S-SOS.

