# OpenReview forum: "S-SOS: Stochastic Sum-Of-Squares for Parametric Polynomial Optimization"
_NeurIPS.cc/2024/Conference — NeurIPS 2024 poster_

### Official Review · Reviewer_iKyk · 2024-07-05

**Soundness:** 3
**Presentation:** 2
**Contribution:** 2
**Rating:** 5
**Confidence:** 3

**Summary:**

This paper is concerned with the following natural generalization of polynomial optizimation:
Given a distribution $\nu(\omega)$ and a polynomial $f(x,\omega)$, find the tightest (in expectation) function $c(\omega)$ that lower bounds $f(x,\omega)$  everywhere.

Given the computational hardness of the problem, the authors focus on the sum-of-squares relaxation (of a given degree $s$).

For trigonometric polynomials, they show a $O(\frac{\log s}{s})$ convergence of the error of the relaxation w.r.t. the optimum (here we hide factors depending on the input in the $O$ notation). Under stronger assumption on the optimal solution they further achieve fast error convergence.
Finally, they complement these results with experimental evaluations (in which they apply certain SDP sparsification techniques to improve the scalability).

**Strengths:**

I find the question  investigated in this paper and the two main theorems (2.1 and 2.2) valuable and non-obvious.

The "Cluster basis hierarchy" is also a non-trivial original contribution.

Finally, I believe the experiments included in the paper will be of interest to the wider NeurIPS' audience.

**Weaknesses:**

In my opinion, the main weakness of the paper lies in the fact that it  fails to explain the  *core* ideas behind its results. There is not a careful discussion of the theorems, their limitations and the insight required to prove them. This makes the paper easy to follow but also quite frustrating to read.  I believe significant improvements can be made in this sense. Such improvements  could greatly improve the quality of the manuscript.

The theoretical analysis result is limited to trigonometric polynomials. This is an important restriction, but a discussion addressing this topic, possible extensions and related obstacles is not present.

The cluster basis hierarchy (henceforth CBH) used to scale up the SDP  is non-trivial and interesting. However it is never discussed in-depth and no theoretical result is provided. I understand that convergence results for CBH may be hard to prove, but a more comprehensive discussion seems needed.

**Questions:**

What can the authors say for other classes of polynomials?

**Limitations:**

I encourage the authors to further discuss the limitations of their theoretical results.

---

> ### Author Rebuttal · Authors · 2024-08-06
>
> We thank Reviewer iKyk for their detailed comments.
>
> We believe that the reviewer’s primary concerns are addressed by our global response:
> - Limitations of trigonometric polynomial assumption and the possibility of extending to more general polynomial families
> - Expanded cluster basis hierarchy discussion
> - Small size of experiments
>
> Regarding the core ideas behind our results and a more careful discussion of the theorems, limitations, and insight - we hope to improve and expand our discussion to make the journey easier and more fruitful.
>
> Thank you again for your consideration.

---

### Official Review · Reviewer_Dyr8 · 2024-07-12

**Soundness:** 3
**Presentation:** 3
**Contribution:** 3
**Rating:** 7
**Confidence:** 4

**Summary:**

The manuscript considers variants of stochastic-SOS hierarchy for parametric polynomial optimization (POP), which had been previously considered using similar (joint+marginal) methods (https://arxiv.org/abs/0905.2497). Unfortunately the (joint+marginal) methods did not work very well, perhaps because parametric POP is known to have a very complicated structure (https://doi.org/10.1287/moor.2021.0097). While the numerical experiments are not exactly conclusive, the present manuscript provides guarantees on the rate of convergence, which improve upon the joint+marginal method somewhat. The "cluster basis" variant of the hierarchy could be of independent interest.

**Strengths:**

-- The convergence rate results are neat, although limited to 1-periodic trigonometric polynomials over compact sets.

-- The manuscript cites a number of interesting recent preprints (such as https://theses.hal.science/LAAS-POP/hal-04201167v1).

**Weaknesses:**

-- The numerical experiments are poorly designed. The sensor network localization problem is known to be reducible (https://arxiv.org/abs/1002.0013) to trivial sizes. Even early papers (https://link.springer.com/article/10.1007/s10107-009-0338-x) considered n=10000 senosrs. The 2010 paper tests on instances with n = 20000 to 100000 sensors, while the present authors consider instances on less than n=15 sensors. This is 4-5 orders of magnitude less than the state of the art.

-- It is not clear how the objective of the heuristically extracted solution (A.7.6) differs from the objective function of the relaxation.

**Questions:**

Could you explain the application of the joint+marginal method as a starting point?

Could you comment on the rates of convergence of the  joint+marginal method?

Could you demonstrate the behaviour of your S-SOS on the more complicated behaviours from https://doi.org/10.1287/moor.2021.0097? Cf. Definition 3.9 (Irregular accumulation point), but also Definition 3.7 (Discontinuous non-isolated multiple point) and Definition 3.6 (Discontinuous isolated multiple point).

**Limitations:**

The discussion of the limitations is misleading. It claims that all SDP-based algorithms fail to scale, while there clearly are (https://arxiv.org/abs/1002.0013) SDP-based algorithms that scale well.

---

> ### Author Rebuttal · Authors · 2024-08-06
>
> We thank reviewer Dyr8 for their detailed comments. In particular, we appreciate all the external citations they flagged.
>
> The reviewer highlights that the sensor network localization (SNL) problem can be reduced to trivial sizes citing work solving instances of 20k-100k sensors, while our numeric experiments only have up to 15 sensors. A few things need to be clarified on this front.
>
> In the cited paper (Krislock and Wolkowicz 2018, https://arxiv.org/abs/1002.0013), the authors propose an algebraic reduction of the noiseless SNL problem so that they can analytically obtain the range of the PSD matrix. This simplifies the SDP dramatically, however in the noisy setting (where observed sensor-sensor distances are perturbed with observation noise) this approach is unusable, requiring significant modification. Krislock and Wolkowicz use this reduction to solve SNL problems of 10k-100k sensors, but they develop a highly specialized algorithm that does not use any SDP solvers, as per the abstract.
>
> Noiseless SNL dramatically simplifies the problem. The intuition here is that in the noiseless setting, localizing even a small number of sensors near an anchor will propagate the correctly localized positions to sensors nearby. If one can find small groups of sensors that can be well-oriented w.r.t. each other, one can pursue localization of these groups in parallel and orient them globally at the very end. In the noisy setting, any error in localization can easily propagate to the whole instance, dramatically reducing the size of problems that can be effectively solved (c.f. https://link.springer.com/article/10.1007/s11276-007-0034-9, https://epubs.siam.org/doi/10.1137/100792366).
>
> The solution extraction in A.7.6 uses the moment matrix to find the mean/variance of the sensor positions. We did not clarify this in sufficient detail in the paper but Lasserre 2010 [[link](https://epubs.siam.org/doi/10.1137/090759240)] discusses and proves the convergence of a sequence of semidefinite relaxations (here, our dual). Our dual (their primal) has moment vector solutions converging to that of a probability distribution encoding all globally optimal solutions. Thus, as long as we solve the SDP with large enough degree $s$ to high accuracy, our extracted solutions will be “close” to the true solution of the infinite-degree relaxation. In our cluster basis hierarchy, we do not prove the convergence of the same moment vector. Numerics seem to suggest that it behaves similarly as the full basis hierarchy, but this is presently a limitation of our theory, and we hope to find a similar convergence proof.
>
> Due to the limited space available, we could not add much detail on the joint+marginal method. The joint+marginal method is essentially our dual semifinite program (our Eq 3 and 4). In that paper, no quantitative convergence rates are provided, although later works (such as the ones we reference) have them.
>
> Thank you for bringing up the Bellon et al paper. In Bellon et al, a parametric SDP has a trajectory of solutions (a sequence of PSD matrices $X$ parameterized by $t$) and seeks to characterize the geometry of the trajectory $(X, t)$. In our work, the primal SDP has as its objective $\int c(\omega) d\nu(\omega)$ where $c(\omega)$ is guaranteed to be a lower-bound to the cost function $f(x, \omega)$ and $\nu(\omega)$ is the density of $\omega$. When we solve the finite-degree primal SDP (Eq 2), we obtain a PSD matrix $W$ that describes the “non-negative part” of the function $f(x, \omega)$ for all $\omega$. As formulated we cannot directly compare our results with that of Bellon et al.
>
> We instead consider the related problems
> $$ \max c s.t. f(x, \omega) - c = m(x)^T W(\omega) m(x), W(\omega) \succcurlyeq 0 $$
> with dual
> $$ \min \langle M(\omega), K \rangle s.t. \langle K_a, M(\omega) \rangle = 0, M(\omega) \succcurlyeq 0 $$
> which describes a series of $(c, W, M)$ values that depend on the parameter $\omega \in \Omega$. We must further restrict to an open interval $\Omega = (\omega_i, \omega_f) \subseteq \mathbb{R}$. Now we have a dual SDP that matches the parametric SDP of Bellon et al.
>
> Let’s first analyze the simple function $f(x, \omega) = (x - \omega)^2 + (\omega x)^2$ with $\omega \in [-1, 1]$. The parametric SDP amounts to finding a SOS approximation to the function at every $\omega$. We can analytically solve for the lower-bounding path $c(\omega)$ and the SOS residual $m(x)^T W(\omega) m(x)$. We find:
> $$ m(x)^T W(\omega) m(x) = f(x, \omega) - c(\omega) = \frac{1}{\omega^2 + 1} (\omega^2 x - \omega + x)^2 $$
> The path $(W, c)$ taken as $\omega$ varies in $[-1, 1]$ sweeps out a smooth curve and we conclude that all points $(W, c)$ are regular for $\omega \in [-1, 1]$.
>
> It is more difficult for SNL, although we have a few observations to start with:
> - In the noiseless setting (parameter $\omega = 0$) there exists a unique configuration of the sensor positions.
> - In the noisy setting, for small $\omega$, the recovered sensor positions will be “close” to the correct unique sensor positions.
> - Consider 3 anchors (A, B, C) and 1 sensor (X) in 2D SNL. If the 3 anchors are not degenerate, then when $\omega=0$ we have the sensor positioned at the intersection of three circles, each having radius equal to the noiseless distance. If we perturb one of the observed distances (X-C) along a line, we note that there are two perturbing values that would lead to loss = 0, leading to consistent distances. This corresponds to the circle intersecting the other two at two points, with mirror symmetry about the segment intersecting A and B.
>
> This last observation helps us see that we should have a discontinuity when the perturbation places the sensor at the segment intersecting A and B, and thus we should have multi-valued $W, M$ on either side of that point.
>
> It’s possible that we misunderstood the work somehow. Regardless, it’s quite interesting and we shall incorporate this into our paper.
>
> Thank you for your consideration.

---

> > ### Comment · Reviewer_Dyr8 · 2024-08-12
> > **Thank you!**
> >
> > Many thanks for having worked out the example. I see the distinction between the separated and non-separated parametric problems, where Bellon considered the former and you are consider the latter. Thank you!

---

### Official Review · Reviewer_Cxce · 2024-07-12

**Soundness:** 3
**Presentation:** 3
**Contribution:** 3
**Rating:** 6
**Confidence:** 2

**Summary:**

This work proposes a new sum of squares (SOS) based relaxation for stochastic polynomial optimization,called Stochastic SOS (S-SOS). For a class of problems where the cost function $f$ can depend on variables $x$ as well as randomly drawn parameters $\omega$, the authors propose a hierarchy of relaxations whose solutions yield strict lower bounds on the optimal value of $f$. For a degree $s$ relaxation, it is shown that the obtained lower bound is $\log(s)/s$ and $1/s^2$ for different parameterizations of the parametric lower bound function $c(\omega)$. Finally, the proposed methods are applied to sensor network localization problems, where S-SOS outperforms the stated baseline.

**Strengths:**

1. This proposes S-SOS, a novel heirarchy of relaxations for solving stochastic polynomial optimization problems.
2. The paper provides asymptotic convergence results for the sequence of relaxations - that is, as the degree of the SOS program increases, the error between the lower bound obtained via the relaxation and the true optimal value reduces to 0. In particular, the finding that by using piecewise constant approximation of $c(\omega)$ achieves the $1/s^2$ convergence rate, outperforming the polynomial parameterization of $c(\omega)$, is quite interesting and surprising to me.
3. The paper is written quite clearly, with no major typos that I could find.
4. The proofs appear to be correct, with a couple of small typos such as in equation 6.

**Weaknesses:**

1. In equation (6), I believe the polynomial in question should be $f(x,\omega) = (x-\omega)^2 + (\omega x)^2$. Otherwise, the correct solution should be $c^*(\omega) = -\omega^4/(1-\omega^2)$.
2. Perhaps in the related work section, a little space could be devoted to other approaches to polynomial optimization, such as the LP/SDP relaxations constructed using the Polya and Handelman theorems, and the LP/SOCP relaxations obtained by using DSOS and SDSOS polynomials.

**Questions:**

1. Are there problems for which the exact lower bound can be achieved in a "single step"? That is, suppose the polynomial cost and constraint functions are of degree $s$, are there problems for which a degree $s$ relaxation suffices to obtain the true minimum? Do such problems arise in practical circumstances?

2. While the DSOS/SDSOS hierarchies do not, in general, converge to global optima, have the authors considered using the associated LP relaxations, particularly in the context of the sensor localization problem? This is salient since the LP/SOCP relaxations can be used to solve much larger problems than the SDP solutions.

3. Is there any intuition as to how the result stated in Prop. 2.2. can be extended to the multivariate case?

**Limitations:**

The limitations of the work are discussed, though they aren't stated in a single section. It would be helpful to add such a section, perhaps in the appendix.

---

> ### Author Rebuttal · Authors · 2024-08-06
>
> We thank Reviewer Cxce for their detailed comments.
>
> With regards to the typos identified, we appreciate the flagging of these and will correct them in the next version. The reviewer is indeed correct in that Equation (6) should have the plus sign.
>
> We thank the reviewer for flagging additional approaches to polynomial optimization and will include those citations. In particular, we find the DSOS and SDSOS papers of particular interest. It seems that the core idea is to consider further relaxing the SDP to a SOCP or LP problem, which constrains the PSD cone further. We didn’t consider further relaxations as our goal was to propose a framework using the tightest possible yet still solvable bound (hence the SDP relaxation). We expect that LP and SOCP relaxations would lead to more efficient and scalable algorithms, though at the expense of accuracy.
>
> We observe that in the regular SOS hierarchy, if one starts with a degree-$2s$ SOS polynomial then you will find exact convergence when using the hierarchy at degree-$2s$. This is true in all cases in that setting, but once we pass to the parameterized/stochastic setting of S-SOS, this no longer holds true generically (or even in most practical cases).
>
> Just as an example, consider the case discussed in Section 3.1, where $f(x, \omega) = (x - \omega)^2 + \omega^2 x^2$. The cost is a degree-4 polynomial in two variables but we show no finite-degree polynomial will achieve the exact cost $c^*(\omega) = \omega^4 / (1 + \omega^2)$. Instead, we see that it can be well-approximated by a polynomial of degree $s$ with $s$ small. And indeed, in many practical scenarios, we anticipate that this is the case. Why this is true (that simple polynomials in general have non-polynomial best lower bounds but may be well-approximated by low-degree polynomials nonetheless) is out of scope in our analysis but it is a very interesting practical fact.
>
> The reviewer has a good point that it is possible to extend the idea behind Prop 2.2 to the multivariate case. We neglected this in our work, but it appears that as in the 1D case of Prop 2.2, one can create a piecewise grid approximation to the lower-bounding function and do a SOS decomposition at each point. However, the constants that prefix the rates will get worse and have a dependence on the spatial dimension, i.e. we will have a rate of the form $\frac{C \sqrt{d}}{s_p}$ where $d$ is the dimension of the noise space $\Omega$ and $s_p$ is the number of grid points per dimension. One should also observe that the number of degrees of freedom for this approximation scales exponentially in the space dimension. We shall adapt the paper accordingly.
>
> Thank you for your consideration.

---

> > ### Comment · Reviewer_Dyr8 · 2024-08-10
> > **Huh?**
> >
> > The authors write:
> >
> > **We observe that in the regular SOS hierarchy, if one starts with a degree-2s SOS polynomial then you will find exact convergence when using the hierarchy at degree-2s.**
> >
> > I would suggest that the authors review https://arxiv.org/abs/2403.08329, which analyzes the following:
> > \[
> > \begin{array}{rl}
> > \min_{x \in R} & x \\
> > \mathrm{s.t.} &1-x^2 \geq 0 \textrm{ and }
> >  & x+(1-\varepsilon)x^2 \geq 0 ,\\
> > \end{array}
> > \]
> > parametrized by a scalar $\varepsilon \in [0,1]$. There, the convergence of the moment-SOS hierarchy is finite, but arbitrarily slow as $\varepsilon$ goes to zero.

---

> > > ### Author Response · Authors · 2024-08-12
> > > **Reply to the "Huh?" by Reviewer Dyr8**
> > >
> > > We thank reviewer Dyr8 for their remark. In particular, we appreciate the external citation they flagged.
> > >
> > > Indeed, our comment was not precise and fails to be true in general. To be clearer, we were referring to the unconstrained case $$\min_{x\in\mathbb R^n} p(x)$$ for $p$ being already in sum of squares form. Hence, the observations of https://arxiv.org/abs/2403.08329 do not quite fit to the setting we wanted to stress.
> > >
> > > We hoped for exact convergence with the same degree in the unconstrained setting. This will be true for example in the cases where the objective assumes $0$ as minimum. We would like to point out that the latter is the case for many physical objectives that are in the focus of our research and, thus, we made the claim. One can also observe that if $p(x)-p_{min}$ allows for an SOS representation it will be of same degree as the original objective in the unconstrained case since no cancellation effects can be observed.
> > >
> > > So the remaining question is if we can shift SOS polynomials by constants and stay SOS. This, indeed, seems not to be true in general as the example $$p(x,y,z) = x^2y^2z^2+(x^2-1)^2+(y^2-1)^2+(z^2-1)^2$$ potentially can show (numerically verified). We would like to follow up on this question.
> > >
> > > As a general remark why assuming $\min p =0$ is of interest, we would like to point out that eventually we do not know the exact form of the objective but are given only function values at samples. However, one can assume a physical model that in many cases would allow an exact SOS recovery in one step as the minimum will be zero.
> > >
> > > Furthermore, observe that in the parametric case the situation is more complex.  Here we have to approximate the lower bounding function by polynomials, which in general is a severe restriction and, thus, we cannot expect to obtain finite convergence in any relevant case. Additionally, the degree we use for the SOS hierarchies is lower bounded by the degree we use to approximate the lower bounding function and the question of finite convergence seem less relevant.

---

> ### Comment · Reviewer_Cxce · 2024-08-07
> **Response to Author Rebuttal**
>
> I thank the authors for their thoughtful rebuttal. At this point, I'm happy to maintain my positive score for this work.
>
> **Edit:**
>
> I would like to elaborate on my justifications for keeping the score as is (weak accept): the work, while interesting, (generally) technically correct, and novel, has a few drawbacks that prevent me from granting a higher score. First, the scale of experiments is far lower than previous work, as pointed out by Reviewer Dyr8. Second, the lack of a multivariate bound (which I believe nearly all problems, including the experiments stated here are), makes the analysis a little handwavy (the dependence on the square root of the dimension in a possible multivariate version of Prop. 2.2., as pointed out by the authors, weakens the result slightly from an applicability standpoint).

---

### Official Review · Reviewer_pRMv · 2024-07-13

**Soundness:** 3
**Presentation:** 2
**Contribution:** 2
**Rating:** 3
**Confidence:** 4

**Summary:**

In their study, the authors investigate parametric polynomial optimization where the function to be minimized is \( f(x, \omega) \). Here, \( x \) represents the decision variable, and \( \omega \) signifies a noise parameter. The primary goal is to approximate the best lower bound, \( c^*(\omega) = \inf_x f(x, \omega) \), for each value of \( \omega \). To manage this setup, the Sum-Of-Squares (SOS) hierarchy is adapted, an approach first introduced by Lasserre in his seminal ``Joint and Marginal'' work in the late 2000s.

The paper elaborates on the derivation of this hierarchy and its dual, as well as discussing the rates of convergence under suitable assumptions about the optimal solution \( c^* \), and includes some applications towards the end of the paper. From a technical standpoint, the work seems correct and presents a rigorous approach to handling stochastic variables in polynomial optimization. However, it does not significantly deviate from established methodologies, which may limit its appeal in terms of novelty.

Regarding its relevance to the NeurIPS audience, while the modified hierarchy certainly adds practical value, the paper does not address applications that align closely with the core interests of the community. The experiments, focusing on sensor network localization, seem peripheral to the main areas of interest at NeurIPS, which typically centers around more direct applications to machine learning and artificial intelligence technologies.

In conclusion, although the paper is technically proficient and might captivate a niche audience interested in theoretical optimization, it appears to fall short of the high innovation standards and relevance to ML typically expected for NeurIPS publications.

**Strengths:**

-- Modification of the SOS hierarchy to incorporate noisy parametric problems
-- Error bounds under assumptions

**Weaknesses:**

-- Relevance
-- Novelty

**Questions:**

In your paper, you discuss the adaptation of the Sum-Of-Squares (SOS) hierarchy to handle stochastic parameters in polynomial optimization, which is a significant theoretical advancement. However, the practical applications presented, such as sensor network localization, seem somewhat tangential to the core interests of the NeurIPS community, which often focuses on direct applications in machine learning and artificial intelligence.

Could you elaborate on how the methodologies developed in your study could be applied to more central problems in machine learning? Additionally, are there potential modifications or extensions to the SOS hierarchy that could make it more directly applicable to common challenges in neural network training or optimization under uncertainty?

---

> ### Author Rebuttal · Authors · 2024-08-06
>
> We thank Reviewer pMRv for their detailed comments.
>
> Concerning our work’s novelty and relevance, we believe that our proposal is one of specific interest to the ML/AI community here.
>
> Applied mathematicians have studied SDPs and convex optimization for some time, while ML/AI scientists are more interested in non-convex/scalable methods as they can be applied to deep learning and other topics of recent interest. We believe fruitful work is to be done at the intersection of these two. To give just two examples of papers that fuse ideas across polynomial optimization, SOS methods, and neural networks, consider:
> - Agrawal et al 2019 [[link](http://arxiv.org/abs/1910.12430)]: Introduces differentiable optimization layers in neural networks, which combines ideas from practical convex optimization and produces a new tool to be used in deep learning.
> - Jaini et al 2019 [[link](http://arxiv.org/abs/1905.02325)]: A framework for high-dimensional density estimation inspired by sum-of-squares polynomials.
>
> Our work assumes that both $f(x, \omega)$ and the lower-bounding function $c(\omega)$ are polynomials. Once we characterize how the SOS hierarchy works in this simple case, we can generalize to the cases where both functions are more complex, perhaps even parameterized by neural networks. Insights from understanding the rate of convergence and the speed of solving a given finite-degree SDP can help us design algorithms in the general case where the functions are more complex.
>
> Consider that we propose a general primal/dual program whose solution provides a tight lower bound. To solve it in practice, one may make the polynomial assumption and then truncate at finite degree, which gives rise to the SDPs we analyze here. Another direction may be to solve the entire problem in the neural network setting: parameterize the basis function $m(x, \omega)$ with a neural network, use a smooth interpolating function class for the lower bound $c(\omega)$, and solve the program with an iterative gradient method. Finally, inspired by Agrawal et al 2019, imagine using the lower bound (or extracted solutions) obtained as output from our solved SDP as input to another layer in a neural network.
>
> The reviewer may not be impressed by the size of the problems that can be solved here, but note our discussion on scalability and size in the global response. We will call the reviewer’s attention to the fact that noiseless v. noisy SNL are two very different problems - in the former we can solve SNL instances of ~100k sensors, in the latter it becomes much more difficult. The resulting output of our program actually solves for the global optimum, which gives sensor positions for every possible configuration of parameter $\omega$. Finally, we only use general-purpose SDP solvers in this work. Many additional methods can be explored, such as exploiting more specialized solvers, low-rank structure, sparsity, and subspaces of the PSD cone. We also expect that leveraging GPUs and other methods of acceleration can immensely increase the size and practicality of the framework we propose.
>
> Our work is also relevant to many problems commonly seen in the AI + Science area. Recent works often seek to learn some kind of molecular potential with deep learning, to do something like “deep molecular dynamics”. The goal here is to have some energy function that depends on some external parameters (e.g. noise). We believe that present approaches lack a principled understanding, and that grounding it in a polynomial optimization setting and sum-of-squares methods can lead to generally useful results, if not a new way of thinking about such problems. To such an end, our paper also proposes a cluster basis hierarchy, introduced and used to scale the size of problems we can solve.
>
> Thank you again for your consideration.

---

### Official Review · Reviewer_MyRu · 2024-08-04

**Soundness:** 3
**Presentation:** 2
**Contribution:** 3
**Rating:** 6
**Confidence:** 3

**Summary:**

The Sum of Squares (SOS) technique is well-known in polynomial optimization studies. This paper studies its variant, in which the function to be lower bounded has random parameters drawn from some probability distribution. The main contribution is a cluster-based SDP hierarchy for the Stochastic-SOS (S-SOS) method and a convergence proof for this and the standard Lasserre. Experiments demonstrate the effectiveness of the new hierarchy in the sensor network localization problem (SNL).

**Strengths:**

- One strength lies in the convergence results on S-SOS, which is a vital generalization of the standard SOS. The proof and arguments in the Appendix seem non-trivial and thorough.
- The cluster-basis hierarchy is novel and practically relevant. Although SOS has good theoretical performance, the SDPs could be expensive to solve in real-world engineering problems. The proposed hierarchy reduces the sizes of the SDPs that need to be solved.
- Applications to the SNL problem complement the theoretical results in convergence and demonstrate the advantage of leveraging the S-SOS technique over standard Monte Carlo methods.

**Weaknesses:**

- It would be better if the authors presented the assumptions in a subsection and explained their restrictiveness. For example, Theorem 2.1 assumes the objective polynomial to be trigonometric. I wonder how challenging it is to extend the results to more general polynomial families.

- The paper uses only two paragraphs to introduce and illustrate the cluster-basis hierarchy, a vital part of its contributions. A more comprehensive discussion of the hierarchy could enhance the paper's content and readability. Besides, I would suggest the authors polish the main paper more.

- Novelties in the proof techniques would often attract independent interests. I wonder if the convergence proof is established by applying only the standard techniques and existing results.

- The experiments seem to contain, at most, $N=15$ sensors, which are suitable for synthetic validation but may not be sufficient to demonstrate the technique's practical advantage.

**Questions:**

Please check the comments on the paper's strengths and weaknesses. The authors should also feel free to point out any inaccuracy or misunderstanding. Thanks.

**Limitations:**

It may be beneficial if the authors could explain more about the paper's limitations in theory (restrictiveness of the assumptions) and practice (practicality of the experimental settings). But I do not find the current presentation concerning. Similarly, I do not think the paper would have any negative societal impact.

---

> ### Author Rebuttal · Authors · 2024-08-06
>
> We thank Reviewer MyRu for their detailed comments.
>
> The reviewer brings up a good point about presenting the assumptions in a subsection. We will set aside a section in the appendix for this purpose.
>
> As for novelty in our proof techniques or lack thereof, it was not obvious to us that one could take the existing results and just apply them directly to a new framework. The way a proof is presented also matters and we hope that our detailed presentation in the appendix is instructive for similar paths taken in the future.
>
> We believe our global response covers the remaining points brought up in the strengths/weaknesses sections, which center on the limitations of our theory and numerics:
> - Limitations of trigonometric polynomial assumption and the possibility of extending to more general polynomial families
> - Expanded cluster basis hierarchy discussion
> - Small size of experiments
>
> Thank you again for your consideration.

---

### Author Rebuttal · Authors · 2024-08-06

We saw the following points come up in multiple reviews so we thought it would make sense to address it as a global response.

The reviewers chiefly seemed to have concerns about the following:
- The limitations of our theoretical assumptions (particularly the trigonometric polynomial one) and the possibility of extending to more general polynomial families
- Clarity, motivation, and further discussion (e.g. more on limitations, cluster basis hierarchy)
- The limitations of our numerical experiments, such as the relevance of sensor network localization and the small size of our experiments

**Limitations of theory.**
With respect to our theory, we agree wholeheartedly that the specialization to 1-periodic trigonometric polynomials is a severe limitation. To review, our work proves quantitative convergence of the S-SOS hierarchy in the setting of (i) a compact domain $X, \Omega$ and (ii) 1-periodic trigonometric polynomials. Periodic trigonometric polynomials are a natural choice to take advantage of Fourier convergence results on a compact domain. If the function $f(x, \omega)$ can be assumed to be only of interest for $(x, \omega)$ in some compact set, then we can rescale the compact set to be a 1-periodic compact domain and apply the 1-periodic trigonometric polynomial results. To use other polynomial families, one can apply a substitution argument, i.e. any result for the trigonometric polynomial hierarchy leads to a matching result for regular polynomials (2.2 in Bach Rudi 2023, link). We regret that we didn’t mention this and will add this to our paper.

**Improved clarity and discussion, esp. on the cluster basis hierarchy.**
The reviewers also mentioned that our paper could have been written more clearly and with much more discussion and elaboration, including on the motivations and intuitions behind the theory as well as the cluster basis hierarchy. Due to the limited space of the Neurips venue, we squeezed what we could into the main text and put everything else in the appendix. We hope to greatly expand the supplement as we continue our revisions.

As for the cluster basis hierarchy, we agree that more discussion is needed — as it is a core technique behind our goal of scaling up the S-SOS approach, particularly when the cost function is well-structured. We will expand our discussion of it in the supplement and hope to cover it in more detail in future work. Unfortunately, it is difficult to prove similar theoretical results for the cluster basis. But here we can also provide numerical support for the convergence of the cluster basis.

**Limitations of numerics: relevance and small scale.**
Finally, with respect to our numerics, the reviewers commented that the core problem (sensor network localization, SNL) is interesting but not of relevance to the broader Neurips community and we only demonstrated it on $N=15$ sensors, a small number compared to the state-of-the-art. We commented in response to another reviewer but will reproduce our response here.

*In the cited paper (Krislock and Wolkowicz 2018, https://arxiv.org/abs/1002.0013), the authors propose an algebraic reduction of the noiseless SNL problem so that they can analytically obtain the range of the PSD matrix. This simplifies the SDP dramatically, however in the noisy setting (where observed sensor-sensor distances are perturbed with observation noise) this approach is unusable, requiring significant modification. Krislock and Wolkowicz use this reduction to solve SNL problems of 10k-100k sensors, but they develop a highly specialized algorithm that does not use any SDP solvers, as per the abstract.*

*Noiseless SNL dramatically simplifies the problem. The intuition here being that in the noiseless setting, localizing even a small number of sensors near an anchor will propagate the correctly localized positions to sensors nearby. As such, if one can find small groups of sensors that can be well-oriented with respect to each other, one can pursue localization of these groups in parallel and then orient them globally at the very end. In the noisy setting, any error in localization can easily propagate to the whole instance, dramatically reducing the size of problems that can be effectively solved (c.f. https://link.springer.com/article/10.1007/s11276-007-0034-9, https://epubs.siam.org/doi/10.1137/100792366).*

Note also that the $N=15$ number is deceptive. **Uncertainty makes the SNL problem considerably more difficult than the noiseless setting.** S-SOS handles this naturally while also solving the problem for its global optimum. This means that we get a solution for the sensor positions for every possible configuration of noise, via the probability distribution $\mu(x, \omega)$.

As for the relevance to Neurips: sensor network localization is a problem that this conference is quite unfamiliar with. It is an old problem in polynomial optimization but generally challenging in the noisy setting. Once lifted into the stochastic setting, we find that our S-SOS framework is a natural fit for this but also many other possible problems in "AI + Science". We believe that taking this approach can be quite fruitful and we hope to see this line of work through.

We want to thank all reviewers for their consideration.

---

### Decision · Program_Chairs · 2024-09-25

**Decision:**

Accept (poster)

**Comment:**

This paper considers an important extension of sum-of-squares for parametric polynomial optimization. In particular, the authors consider its stochastic counterpart and provide quantitative convergence results of the hierarchy as the degree of the polynomial increases. An application to sensor network localization problems is shown.

My own assessment is that the paper is a valuable and unique contribution the NeurIPS community, focusing on core stochastic optimization theory within the important SOS framework. There were some concerns about the size of the numerical experiment, but to the AC, the theoretical findings moved the quality needle across the bar for acceptance.